# Targeting Group 3 Medulloblastoma by the Anti-PRUNE-1 and Anti-LSD1/KDM1A Epigenetic Molecules

**DOI:** 10.3390/ijms25073917

**Published:** 2024-03-31

**Authors:** Francesca Bibbò, Fatemeh Asadzadeh, Angelo Boccia, Carmen Sorice, Orazio Bianco, Carmen Daniela Saccà, Barbara Majello, Vittoria Donofrio, Delfina Bifano, Lucia De Martino, Lucia Quaglietta, Adriana Cristofano, Eugenio Maria Covelli, Giuseppe Cinalli, Veronica Ferrucci, Pasqualino De Antonellis, Massimo Zollo

**Affiliations:** 1Department of Molecular Medicine and Medical Biotechnological DMMBM, University Federico II of Naples, 80131 Naples, Italy; francesca.bibbo@unina.it (F.B.); veronica.ferucci@unina.it (V.F.); pasqualino.deantonellis@unina.it (P.D.A.); 2CEINGE Biotecnologie Avanzate “Franco Salvatore”, 80131 Naples, Italy; asadzadeh@ceinge.unina.it (F.A.); boccia@ceinge.unina.it (A.B.); sorice@ceinge.unina.it (C.S.); bianco@ceinge.unina.it (O.B.); 3SEMM European School of Molecular Medicine, 20139 Milan, Italy; 4Department of Biology, University Federico II of Naples, 80138 Naples, Italy; carmendaniela.sacca@unina.it (C.D.S.); majello@unina.it (B.M.); 5Department of Pathology, Santobono-Pausilipon Children’s Hospital, AORN, 80129 Naples, Italy; v.donofrio@santobonopausilipon.it (V.D.); d.bifano@santobonopausilipon.it (D.B.); 6Pediatric Neuro-Oncology, Santobono-Pausilipon Children’s Hospital, AORN, 80129 Naples, Italy; demartinoluci@gmail.com (L.D.M.); l.quaglietta@santobonopausilipon.it (L.Q.); 7Pediatric Neuroradiology, Santobono-Pausilipon Children’s Hospital, AORN, 80129 Naples, Italy; a.cristofano@santobonopausilipon.it (A.C.); e.covelli@santobonopausilipon.it (E.M.C.); 8Pediatric Neurosurgery, Santobono-Pausilipon Children’s Hospital, AORN, 80129 Naples, Italy; giuseppe.cinalli@gmail.com; 9DAI Medicina di Laboratorio e Trasfusionale, ‘AOU Federico II Policlinico’, 80131 Naples, Italy

**Keywords:** medulloblastoma, Prune-1, KDM1A, TGF-β, metastasis, epigenetics, immune system

## Abstract

Medulloblastoma (MB) is a highly malignant childhood brain tumor. Group 3 MB (Gr3 MB) is considered to have the most metastatic potential, and tailored therapies for Gr3 MB are currently lacking. Gr3 MB is driven by PRUNE-1 amplification or overexpression. In this paper, we found that PRUNE-1 was transcriptionally regulated by lysine demethylase LSD1/KDM1A. This study aimed to investigate the therapeutic potential of inhibiting both PRUNE-1 and LSD1/KDM1A with the selective inhibitors AA7.1 and SP-2577, respectively. We found that the pharmacological inhibition had a substantial efficacy on targeting the metastatic axis driven by PRUNE-1 (PRUNE-1-OTX2-TGFβ-PTEN) in Gr3 MB. Using RNA seq transcriptomic feature data in Gr3 MB primary cells, we provide evidence that the combination of AA7.1 and SP-2577 positively affects neuronal commitment, confirmed by glial fibrillary acidic protein (GFAP)-positive differentiation and the inhibition of the cytotoxic components of the tumor microenvironment and the epithelial–mesenchymal transition (EMT) by the down-regulation of N-Cadherin protein expression. We also identified an impairing action on the mitochondrial metabolism and, consequently, oxidative phosphorylation, thus depriving tumors cells of an important source of energy. Furthermore, by overlapping the genomic mutational signatures through WES sequence analyses with RNA seq transcriptomic feature data, we propose in this paper that the combination of these two small molecules can be used in a second-line treatment in advanced therapeutics against Gr3 MB. Our study demonstrates that the usage of PRUNE-1 and LSD1/KDM1A inhibitors in combination represents a novel therapeutic approach for these highly aggressive metastatic MB tumors.

## 1. Introduction

Medulloblastoma (MB) is a devastating cerebellar tumor that accounts for nearly ~20% of all primary CNS tumors in children under 14 years of age [1]. MB is classified into four subgroups with different molecular and clinical features [2,3,4,5,6]. An in-depth classification of MB, obtained by overlapping genetic and epigenetic features, consists of twelve subtypes [7]: two WNT (α and β); four SHH (α, β, γ, and δ); three Gr3 (α, β, and γ); and three Gr4 (α, β, and γ). WNT MB has a favorable outcome and very low frequency of metastatic disease, while Gr3 MB and Gr4 MB are considered to have a high metastatic potential [7].

Of interest, genomic profiling studies have recently reported a high heterogeneity in Gr3 and Gr4 MB in terms of molecular and clinical features, but also the existence of a subset of tumors with overlapping signatures [8]. The impact of this genomic profiling on the patients’ therapy has been limited to trials testing smoothened (SMO) antagonists for patients with SHH MB [9] and efforts to reduce therapy for children affected by WNT MB [10]. Tailored therapies for Gr3 MB are currently lacking, and the combination of surgery, craniospinal radiotherapy (except in young children for whom radiotherapy has devastating neurocognitive side effects), and multi-agent chemotherapy is used [1]. The current clinical protocol for MB is SIOP PNET5. The classical first element of therapy is maximal safe tumor resection to reduce the tumor volume as much as possible [11]. Radiotherapy in SIOP PNET5 MB should start within 28 days after first tumor surgery is delivered to the entire central nervous system as craniospinal irradiation (CSI), followed by a boost to the tumor region, in once daily, five weekly fractions of 1.8 Gy. The CSI dose depends on the risk stratum [12,13]. For the maintenance chemotherapy in SIOP PNET5 MB, the ‘B’-block does not contain cisplatin but just vincristine and cyclophosphamide. The duration of maintenance chemotherapy in SIOP PNET5 MB depends on the stratum [13]. Although, recently, Michalski et al. [14] carried out a phase III trial of reduced-dose and reduced-volume radiotherapy with chemotherapy for newly diagnosed average-risk MB [14], the molecular drivers of metastatic dissemination in MB remain elusive. Recently, we highlighted a new metastatic axis (independent of c-MYC amplification) in Gr3 MB. In Gr3, we found that Prune-1 enhances the canonical TGF-β pathway followed by OTX2 and SNAIL up-regulation and PTEN down-regulation [15]. Furthermore, OTX2 maintains rhombic lip (RL) identity by inhibiting the CBFA complex until the cells exit the RL and differentiate [16]. The overexpression of OTX2 results in failed RL differentiation. The resulting ball of RL progenitor cells are retained with the RL in the nascent nodulus of the developing cerebellum, where ongoing mitotic activity eventually results in a mass lesion diagnosed as Gr3 or Gr4 MB [16]. Thus, OTX2 could be an important target for future therapies of MB.

Prune-1 is a member of the DHH (Asp-His-His) protein superfamily, with exopolyphosphatase activity for short-chain over long-chain inorganic polyphosphates [17]. Prune-1 is highly expressed in the central nervous system during embryonic and fetal development and plays a crucial role in cell migration and proliferation, interacting with proteins involved in cytoskeletal rearrangement [18,19,20]. Patients with loss of exopolyphosphatase PPX/PPASE enzymatic activity of Prune-1 showed delayed myelination, thin corpus callosum, white matter abnormalities, mild frontal cerebral atrophy, and prominent cerebellar atrophy [21], demonstrating the importance of its correct dosage for brain pathophysiology. Its overexpression promotes cell motility and cancer progression and metastasis [22,23]. Prune-1 is a multi-domain adaptor protein with an unfolded domain, allowing interactions with several binding partners and the modulation of different signaling cascades, including WNT and TGF-β signaling, and serves as nodal points in the regulation of many cellular activities [24,25] The WNT and TGF-β pathways are cell–cell signaling systems that control a plethora of processes, from embryonic development and cell proliferation, differentiation and migration, to tissue homeostasis, stem cell behavior, tissue regeneration, and cancers [26]. TGF-β is also known to regulate the systemic immune surveillance of the tumor host by controlling immune responses, and it maintains immune homeostasis through its impact on proliferation, differentiation, and in the tumor microenvironment (TME) of multiple immune cell lineages [27].

Recently, a pyrimido-pyrimidine derivative (AA7.1) was found with the ability to decrease Prune-1 protein intracellular levels via the enhancement in its degradation [15]. This molecule was also found able to decrease Prune-1 mRNA and protein levels in different Gr3 MB [15] and triple-negative breast cancer (TNBC) [28] cells. In vitro and in vivo assays also showed the ability of AA7.1 to decrease the cell proliferative and migratory processes, impairing PRUNE-1- and NME1-related signaling and influencing cancer energy metabolism, the TME, and metastasis [25]. In detail, in Gr3 MB, AA7.1 impaired the metastatic axis driven by Prune-1, thus leading to impairment in TGF-β, decreased levels of OTX2, the up-regulation of PTEN, the inhibition of the epithelial to mesenchymal transition (EMT), a reduction in Nestin, and increases in Tuj1 and GFAP differentiation neuronal markers [15]. The pharmacological inhibition of Prune-1 protein in TNBC (achieved through AA7.1 treatment) decreased the number of metastatic foci in vivo also via inhibiting the switch of tumor-associated macrophages (TAMs) in the TME toward the M2 phenotype [28]. In TNBC, AA7.1-mediated tumor inhibition occurred through the impairment in the TGF-β pathway, reduction in inflammatory cytokines (IL-17F), and the modulation of the protein content of extracellular vesicles, including vimentin [28]. Importantly, this small molecule is not toxic [15]. Thus, altogether, these findings suggest that AA7.1 is a non-toxic potential immunomodulatory molecule with the ability to modulate the inflammatory processes in the TME, thus inhibiting the metastatic spread in Prune-1-overexpressing tumors.

In MB, an additional level of regulation is mediated by epigenetics, consisting of large hypomethylated chromosomal regions that cause increased gene expression [8,29]. The epigenetic regulators that mostly affect MB are DNA methylation and histone modifications [30]. Lysine demethylase 1 (LSD1) is the most important epigenetic modulator. LSD1, encoded by the KDM1A gene, is a member of the flavine adenine dinucleotide-dependent (FAD-dependent) amine oxidase (AO) family of demethylases, integrated in several chromatin-modifying multiprotein complexes, such as rest co-repressor (CoREST) and nucleosome remodeling and deacetylase (NuRD) [31,32,33]. It is known to demethylate lysines 4 and 9 on the tail of histone 3 [34,35]. When methyl groups are removed from mono- and di-methylated H3K4, LSD1/KDM1A is known to repress gene expression [34]. Conversely, it has been reported to activate gene transcription by demethylating H3K9 in complex with nuclear hormone receptors [35]. LSD1n, a neuronal-specific isoform, demethylates the transcription mark H4K20me1 [36]. Histone H4 lysine 20 mono-methylation facilitates chromatin openness [37]. LSD1/KDM1A is overexpressed in many proliferative diseases, including cancers [38], highlighting its use as a potential target in oncology. In addition, since LSD1/KDM1A is recruited to target genes via interaction with transcription factors, the catalysis-independent protein–protein interaction (PPI) with SNAIL/growth factor independent 1 (SNAG)-domain transcription factors (TFs) has been found to be essential for LSD1/KDM1A as a cancer driver [39,40].

In MB, Pajtler et al. [41] found that the histone demethylase LSD1/KDM1A is functionally involved in the regulation of the malignant phenotype of medulloblastoma cells by influencing three major hallmarks of cancer cells, uncontrolled cell proliferation, the avoidance of apoptosis, and migratory capacity. The treatment of medulloblastoma cells with a novel specific LSD1/KDM1A inhibitor, the small molecule NCL-1, led to the significant inhibition of cellular growth in vitro [41]. In another report from Callegari et al. [42], a strong correlation between the high expression of LSD1/KDM1A and poor survival in patients with SHH-δ tumors was observed [42]. Moreover, Lee et al. [42] showed that LSD1/KDM1A is essential for Gfi1-mediated transformation in MB and that the pharmacological inhibitors of LSD1/KDM1A potently inhibit the growth of Gfi1-driven tumors [43].

Several molecules have been developed to date to target LSD1/KDM1A. A host of classical amine oxidase inhibitors, such as tranylcypromine (TCP), pargyline, and phenelzine, together with LSD1 tool compounds, such as SP-2509 and GSK-LSD1, have been extensively utilized in LSD1 mechanistic cancer studies. Several optimized new chemical entities have reached clinical trials in oncology, such as ORY-1001 (Iadademstat), GSK2879552, SP-2577 (Seclidemstat), IMG-7289 (Bomedemstat), INCB059872, and CC-90011 (Pulrodemstat) [44]. To date, several TCP-based LSD1/KDM1A inhibitors have entered clinical trials for the treatment of AML and SCLC [45,46]. Also, the reversible LSD1 inhibitors CC-90011 and SP-2577 have recently entered clinical trials [46,47].

Seclidemstat (SP-2577) is an inhibitor of LSD1/KDM1A, currently being evaluated in combination with AZA for MDS/CMML, following HMA-failure in a phase 1/2 study (NCT04734990) [44].

In this paper, we investigated the role of the PRUNE-1 inhibitor “AA7.1” in combination with the LSD1/KDM1A inhibitor (Seclidemstat) in Gr3 MB cells. Although the targeting of epigenetic modulators is an increasingly used approach to treat several types of cancers, not much is known about their action and efficacy in pediatric brain cancer, such as MB. Being both drugs targeting TGF-β signaling, we wondered if a combinatorial action would benefit the treatment of the Gr3 MB model presented in this paper. To address this gap in knowledge, we analyzed the effect of the combination of these two drugs on the metastasis axis driven by Prune-1 in Gr3 MB. We identified a significant efficacy on the targets that we had previously identified and which we fully characterized. Finally, we observed that the identified signatures were associated with a lower neuronal commitment and immune suppression, conditions that we found to be reversed in our preliminary study with the use of the AA7.1 and SP-2577 drugs combined. We also observed an important impairment in mitochondrial energy production in tumor cells when the combination was used. The results presented in this paper open the door to future applications in clinics.

## 2. Results

### 2.1. LSD1/KDM1A Expression and Epigenetically Regulation of PRUNE-1 in Medulloblastoma

To investigate the role of LSD1/KDM1A in MB, we analyzed its expression levels by comparing MB and the normal cerebellum. There was a significantly higher LSD1/KDM1A expression in MB (*p* = 7.9 e−20; Figure 1A). Among the MB molecular subgroups, LSD1/KDM1A transcriptional levels were higher in WNT, SHH, and Gr3 MB in the public Cavalli dataset (*p* = 8.56 e−07; Figure 1B).

We then evaluated the prognostic value of LSD1/KDM1A expression in MB patients by a Kaplan–Meyer analysis. The data revealed that LSD1/KDM1A expression was negatively correlated with the survival rate in MB (High = 305; Low = 307; *p* = 0.066; Figure 1C). In the WNT and SHH MB subgroups, the percentage of survival was worse when LSD1/KDM1A had a low expression (Appendix A). Conversely, in Group 4, the percentage of survival was worse when the expression of LSD1/KDM1A was high, without reaching statistical significance (Appendix A). Importantly, this inverse correlation was stronger in Gr3 MB (High = 56; Low = 57; *p* = 1.5 e−03; Figure 1D).

Prune-1 is already known to promote cell migration and metastasis [20,23] and to be highly expressed in MB samples compared to the normal cerebellum [15]. Among the MB molecular subgroups, PRUNE-1 transcriptional levels were significantly higher in Gr3 and Gr4 MB compared to WNT and SHH MB in the public Cavalli dataset [15].

Since LSD1/KDM1A is known to be a transcriptional regulator, we searched for the potential regulation of PRUNE-1 expression. To this aim, we investigated their potential correlation in MB (Cavalli; *n* = 763; R = 0.030; *p* = ns) (Pfister; *n* = 223; R= 0.358; *p* < 0.001) (Figure 1E,G) and within each subgroup (Appendix A–F). We analyzed, especially, the data of Gr3 MB samples by using two different datasets: the Cavalli dataset (*n* = 144; R = 0.168; *p* = 0.044; Figure 1F) and Pfister dataset (*n* = 56; R D425-Med cells treated for= 0.416; *p* = 3.79 e−08; Figure 1H). The data showed a positive correlation. Thus, we hypothesized that LSD1/KDM1A is an epigenetic regulator that can control PRUNE-1 expression.

Next, we observed that the Kaplan–Meyer survival curves for both PRUNE1 and LSD1/KDM1A showed, in Gr3 MB, a similar trend (Figure 1I). For this reason, we investigated whether a higher expression for both genes might have a better prognostic value. We asked whether the combined expression of PRUNE-1 and LSD1/KDM1A could better stratify high-risk MB patients. Of interest, we found that, when both PRUNE-1 and LSD1/KDM1A were highly expressed, the percentage of survival was lower in MB (*n* = 606; *p* = 0.248; Figure 1J) and, in particular, in Gr3 MB (Low/Low = 21; High/High = 34; *p* = 0.0213; Figure 1K).

To further investigate if PRUNE-1 can be epigenetically regulated by LSD1/KDM1A, we initially used genome views of KDM1A ChIP-Seq signals over the PRUNE-1 locus using publicly deposited data [48] (Appendix A). The normalized LSD1/KDM1A signals were visualized with the UCSC Genome Browser, together with ChIP track. The results of this analysis indicate that LSD1/KDM1A peaks show preferential binding at the transcription start site (TSS) of the PRUNE-1 gene promoter in the human neuroblastoma cell line SH-SY5Y [48], a known neural model of differentiation. Because these data belong to an experiment where LSD1/KDM1A was found to be in correlation with the methylation of H3K9, we hypothesized that the presence of LSD1/KDM1A in those chromatin regions leads to the demethylation of H3K9, thus favoring transcriptional activity, suggesting LSD1/KDM1A as a positive epigenetic regulator of PRUNE-1 expression. Of interest, Lee et al. [43] found additional LSD1/KDM1A peaks on PRUNE-1 TSS in murine MB samples (Appendix A).

Altogether, these data indicate that MB, particularly Gr3 MB, shows correlations with high levels of PRUNE-1 and LSD1/KDM1A and that LSD1/KDM1A is a putative epigenetic regulator of PRUNE-1 expression.

### 2.2. The Combination of the Prune-1 Inhibitor AA7.1 and the Selective LSD1/KDM1A Inhibitor SP-2577 impairs Cell Proliferation and TGF-β Signalling in D425-Med Gr3 MB Cells

Following these observations, we decided to test PRUNE-1 expression after the pharmacological inhibition of LSD1/KDM1A by using a Tranilcipromide (TCP) inhibitor, one of the first LSD1/KDM1A inhibitors identified [49], and the novel selective and reversible inhibitor SP-2577 (Seclidemstat). We first assessed the cytotoxicity of the compound TCP in the D425-Med Gr3 MB cell line regarding cell proliferation. The cell index proliferation assay was used to determine the half-maximal inhibitory concentration (IC_50_) of TCP. We tested escalating doses from 0.5 mM to 2 mM of TCP on D425-Med cells, and we found that 1 mM was the IC_50_ after 24 h of treatment (IC_50_ value: 1 mM, R^2^: 9.7739 e−1; Appendix A). Thus, we performed a real-time PCR by using (Sybr Green) on D425-Med cells and D283-Med cells treated with a concentration of 1 mM of TCP. Vehicle-treated cells were used as the control. The results show a down-regulation in the mRNA level of PRUNE-1 when the TCP inhibitor is used (Appendix A). Thus, we focused on a more specific inhibition of LSD1/KDM1A with its selective inhibitor SP-2577. We again performed a cell index proliferation assay to determine the half-maximal inhibitory concentration (IC_50_) of SP-2577. We tested escalating doses from 37.5 µM to 1200 µM of SP-2577 on D425-Med cells, and we found that 50 µM is the IC_50_ after 24 h of treatment (IC_50_ values: 50 µM, R^2^ 9.3894 e−1; Appendix A). We tested also escalating doses from 1 µM to 20 µM of SP-2577 on D283-Med Gr3/4 cells, and we found that 10 µM is the IC_50_ after 24 h of treatment (IC_50_ values: 50, R^2^ 9.0000 e−1; Appendix A). Thus, we performed a real-time PCR by using (Sybr Green) on D425-Med cells treated with a concentration lower than half of the IC_50_ (22.5 µM) of the LSD1/KDM1A inhibitor SP-2577. Vehicle-treated cells were used as the control. The results show a down-regulation in the expression level of PRUNE-1 and OTX2 when LSD1/KDM1A is inhibited by the selective inhibitor (Figure 2A), thus demonstrating that LSD1/KDM1A can enhance PRUNE-1 transcription. The effect on OTX2 transcription may be due to the downregulation of PRUNE-1, to a direct interaction of LSD1/KDM1A with regulatory regions of this gene (as demonstrated by the ChIP-Seq data publicly available (Appendix A), or to both. We hypothesized a potential role of PRUNE-1 and LSD1/KDM1A in MB and how their inhibition may have a therapeutic potential.

Recently, the high expression of Prune-1 was found to be correlated to the activation of the canonical TGF-β pathway through SMAD2/3 phosphorylation, OTX2 and SNAIL upregulation, and PTEN inhibition [14]. So, we hypothesized that, if inhibited by its selective inhibitor SP-2577 (Seclidemstat), LSD1/KDM1A is not able to activate PRUNE-1 transcription and TGF-β pathway. Consequently, if the LSD1/KDM1A inhibitor is used in combination with the PRUNE-1 inhibitor AA7.1, the levels of Prune-1 should be lower with a stronger inhibition of the TGF-β pathway. Thus, we used a concentration lower than half of the IC_50_ (22.5 µM) of SP-2577 to test the cell proliferation in D425-Med when this compound is used in combination with the PRUNE-1 inhibitor AA7.1. Of interest, we found that the combination of SP-2577 (22.5 µM) and AA7.1 (100 µM) enhanced the inhibition of cell proliferation in vitro in D425-Med cells (Figure 2B) and D283-Med Gr3/4 MB cells (Figure 2C) compared to their single usage. Thus, this suggests a synergistic action to inhibit MB proliferation in vitro.

Of importance, the treatment with AA7.1, SP-2577, or with their combination did not contribute to cell death through the activation of the pathway of caspase-3 in D425-Med cells, as demonstrated by the Western Blot analysis of the cleaved caspase-3 (Figure 2D,E).

To investigate the effects of the combination of these compounds on the Prune-1 protein and its downstream targets—TGF-β signaling—we performed an immunoblotting analysis on D425-Med cells treated for 24 h with 22.5 µM of SP-2577 and 100 µM of AA7.1. We observed a greater reduction in the levels of OTX2 and phosphorylated SMAD2 (S467) when the cells were treated with the combination of the two drugs compared to the single usage (Figure 2D,E). Interestingly, when both SP-2577 and AA7.1 were used, the levels of PTEN were increased (Figure 2D,E). Of interest, we observed that the treatment with these drugs and, with their combination, resulted in a decreased protein level of the epithelial–mesenchymal transition (EMT) N-cadherin marker (Figure 2D,E). These data suggest the capability of the combination of these compounds (SP-2577 and AA7.1) to impair TGF-β signaling and to restore the activation of the tumor suppressor PTEN. Thus, we tested the effect of the combination of AA7.1 and SP-2577 on two additional MB cell lines derived from different tumor subgroups (i.e., D283-Med, a Gr3/4 MB cell line, and Daoy, a SHH MB cell line), and we confirmed the reduction in the Prune-1 protein upon the use of both inhibitors as a treatment (see Appendix A), although not statistically relevant, and postulate at this time some additional level of regulation coordinated by these two targets in SHH and Gr4 MB. We wanted also to test the efficacy of the combination of AA7.1 and SP-2577 on primary Gr3 MB cells.

### 2.3. Treatment of Primary Gr3 MB Cells with the Combination of AA7.1 and SP-2577 Confirms its Efficacy on Targets

To confirm the efficacy of the small molecule AA7.1 (PRUNE-1 inhibitor) in combination with the LSD1/KDM1A inhibitor SP-2577 on the Gr3 MB metastatic axis, we performed experiments on primary Gr3 MB cells. We obtained these cells from a biopsy sample derived from neurosurgery resection in a patient affected by Gr3 MB (Figure 3A,B). The images show a posterior fossa mass occupying the fourth ventricle and the left foramen of Luschka. The intraventricular tumor is well-defined, has a heterogeneous structure (largely solid), and is characterized by peripheral necrotic-cyst components, areas of low ADC value, hemorrhagic foci, and signs of mineralization. The compression and displacement of the neighbor brain structure are present, leading to tonsillar herniation, obstructive hydrocephalus, and intracranial hypertension (Figure 3A). The immunohistochemical characterization of tumor tissue revealed the presence of typical syncytial arrangement of undifferentiated cells, poorly differentiated embryonal cells with hyperchromatic and variably shaped nuclei and diffused expression of synaptophysin (Figure 3B). The patient underwent neurosurgery with endoscopic third ventriculostomy (ETV) and the complete removal of the expansive process of the fourth ventricle via a midline suboccipital approach. We performed a whole exome sequencing (WES) analysis on the DNA extracted from this bioptic tumor sample to identify driver mutations. The results obtained by the mutational WES analysis of our Gr3 MB tumor, from which the primary cells were derived, show the presence of three pathogenic variants in the following genes: DAPK1 (NM_001288729; exon 11; c.C938A; p.S313X; stopgain); HSPH1 (NM_001286503; exon 2; c.G124T; p.G42X; stopgain); and LRP1B (NM_018557; exon 25; c.G4168T; p.G1390X; stopgain) (see Table 1). In addition, 56 potentially pathogenic variants were identified (Appendix A). To confirm the data obtained from the D425-Med cells on the efficacy of our drugs to target the axis driven by Prune-1, we treated primary Gr3 MB cells with 100 µM of AA7.1 and 22.5 µM of SP-2577 and we performed Western blotting and RNA-Seq analyses (Figure 3C). We observed, again, a reduction in the quantity of Prune-1 and OTX2 proteins and phosphorylated SMAD3 (S423-425), with stronger effects when the cells were treated with the combination of the two drugs compared to the single usage (Figure 3D,E). We found PTEN levels increased in the treated cells compared to the vehicle cells (Figure 3D,E). These data suggest the ability of the combination of these compounds (SP-2577 and AA7.1) to impair the Prune-1 axis and TGF-β signaling also on primary Gr3 MB cells. Thus, we proceeded with the RNA-seq analysis of the cells treated with AA7.1, SP-2577, and the combination of the two in comparison to the vehicle cells. The analysis of the RNA-seq data on primary Gr3 MB cells showed very interesting differentially expressed genes with the combination of the AA7.1 and SP-2577 drugs compared to the vehicle cells (Figure 4A).

### 2.4. The Combination of AA7.1 and SP-2577 Affects Neuronal Commitment

To better understand the mechanisms of action of these two molecules and to investigate the genes and pathways affected, we treated primary Gr3 MB cells for 24 h with AA7.1 (100 μM), SP-2577 (22.5 μM), and their combination, and we performed an RNA-Seq analysis. Vehicle-treated cells were used as the control. Gene expression analysis showed up-regulation of genes implicated in neuronal differentiation in cells treated with Prune-1 and LSD1/KDM1A inhibitors, with stronger evidence in cells treated with the combination of the two. Of importance, in the cells treated with the combination of the inhibitors up-regulated genes (Appendix A) were observed that are implicated in the following pathways: brain development (GO:0007420; FDR 0.00001); cell morphogenesis involved in neuronal differentiation (GO:0048667; FDR 0.000002); positive regulation of nervous system development (GO:0051962; FDR 0.015); nervous system development (GO:0007399; FDR 0.00000009); neurogenesis (GO:0022008; FDR 0.0001); neuron projection guidance (GO:0097485; FDR 0.02); neuron projection morphogenesis (GO:0048812; FDR 0.000004); axon guidance (GO:0007411; FDR 0.02); axon genesis (GO:0007409; FDR 0.0001); axon development (GO:0061564; FDR 0.00008); synapse assembly (GO:0007416; FDR 0.006); synapse organization (GO:0050808; FDR 0.000002); synaptic signaling (GO:0099536; FDR 0.000002); trans-synaptic signaling (GO:0099537; FDR 0.000009); and chemical synaptic transmission (GO:0007268; FDR 0.00001) (Figure 4B,C). Thus, these results indicate a positive effect of the combination of our two drugs on the neuronal commitment of MB cells. We previously observed that the treatment with AA7.1 resulted in the increased expression of the neuronal differentiation glial fibrillary acidic protein (GFAP) in Gr3 MB xenografted mice [15]. In this paper, to investigate the direct cytological evidence of the treatment on differentiation, we performed immunofluorescence staining of D425-Med cells treated with AA7.1 in combination with SP-2577 using markers of neuronal differentiation. The data presented in Figure 4D show a significant increase in the signal of the GFAP when the cells are treated with the combination of the drugs, demonstrating a commitment toward a more differentiated phenotype (Figure 4D).

We observed also important effects on the genes that may control the phenotype of the components of the TME and genes related to the epithelial–mesenchymal transition (EMT). In this paper, we measured the expression of N-Cadherin protein upon treatment with both inhibitors in combination and alone in D425-Med MB cell lines (see Appendix A), showing a substantial downregulation of the marker upon both treatments. Taken together, these data confirm the RNAseq data showing the combinatorial skill of the inhibitors to induce both differentiation and a reduction in the EMT, of great importance for impairing aggressiveness and metastatic processes in vitro.

### 2.5. The Combination of AA7.1 and SP-2577 Affects Leukocyte Differentiation

Of importance, in primary MB cells treated with the combination of the inhibitors AA7.1 and SP-2577, we found that up-regulated genes (Appendix A) were implicated in the following pathways concerning immune responses: myeloid cell differentiation (GO:0030099; FDR 0.0008); leukocyte differentiation (GO:0002521; FDR 0.00007); regulation of leukocyte differentiation (GO:1902105; FDR 0.03); leukocyte activation (GO:0045321; FDR 0.0002); leukocyte cell–cell adhesion (GO:0007159; FDR 0.005); regulation of leukocyte cell–cell adhesion (GO:1903037; FDR 0.01); antigen processing and presentation of endogenous peptide antigens via MHC class I via the ER pathway, TAP independence (GO:0002486; FDR 0.0004); antigen processing and presentation of peptide antigens via MHC class Ib (GO:0002428; FDR 0.002); positive regulation of lymphocyte activation (GO:0051251; FDR 0.015); lymphocyte activation (GO:0046649; FDR 0.02); positive regulation of T-cell activation (GO:0050870; FDR 0.04); and T-cell activation (GO:0042110; FDR 0.02) (Figure 4B,C). Thus, these results indicate a positive effect of the combination of our two drugs on the expression of genes that regulate the switch of the components of the TME to an immune-responsive and cytotoxic phenotype mostly triggered by TGF-β pathway inhibition. 

Due to the importance of energy production for tumor cells, we asked if the combination of AA7.1 and SP-2577 could also have some effects on the metabolism.

### 2.6. The Combination of AA7.1 and SP-2577 Affects Mitochondrial Metabolism

We investigated the pathways down-regulated by the combination of our two drugs and performed an RNA-Seq analysis. The gene expression analysis showed, in primary Gr3 MB cells treated for 24 h with 100 μM AA7.1 and 22.5 μM SP-2577, a significantly lower expression of several genes (Appendix A) implicated in mitochondrial metabolism, in particular oxidative phosphorylation (OXPHOS), compared to the vehicle cells. Of importance, the treatment with the combination of the inhibitors resulted in lower RNA levels of genes that encode for subunits of mitochondrial respiratory chain proteins: mitochondrial gene expression (GO:0140053; FDR 1.46 e−15); mitochondrial RNA metabolic process (GO:0000959; FDR 0.003); mitochondrial translation (GO:0032543; FDR 1.69 e−13); protein localization to the mitochondrion (GO:0070585; FDR 0.008); mitochondrion organization (GO:0007005; FDR 2.04 e−10); inner mitochondrial membrane organization (GO:0007007; FDR 0.04); mitochondrial respiratory chain complex assembly (GO:0033108; FDR 2.33 e−08); mitochondrial respiratory chain complex I assembly (GO:0032981; FDR 8.77 e−08); NADH dehydrogenase complex assembly (GO:0010257; FDR 8.77 e−07); respiratory electron transport chain (GO:0022904; FDR 0.00003); proton motive force-driven mitochondrial ATP synthesis (GO:0042776; FDR 0.000002); ATP biosynthetic process (GO:0006754; FDR 0.00001); ATP metabolic process (GO:0046034; FDR 0.0004); oxidative phosphorylation (GO:0006119; FDR 0.00009); cellular respiration (GO:0045333; FDR 6.60 e−12); and aerobic respiration (GO:0009060; FDR 1.52 e−08) (Figure 5A,B). Thus, these results indicate that the combination of our two drugs is also able to impair the mitochondrial energy production of tumor MB cells in a very efficient way.

## 3. Discussion

The key role of LSD1/KDM1A in development has been widely recognized [50]. Specific isoforms of LSD1/KDM1A have been described to regulate the transcription of genes in neurons [36,48], suggesting a potential role in neuronal differentiation. A robust reduction in LSD1/KDM1A protein levels was observed in vitro when PC12 cells were differentiated to neurons and cortical neurons [51]. Moreover, LSD1/KDM1A was found physically associated with Gfi1 and helped to jointly inhibit genes involved in neuronal commitment and differentiation [43]. Neuronal commitment and differentiation pathways are down-regulated in ^Myc+Gfi1^MB tumors [43]. Thymidine incorporation assays on cells treated with two different LSD1 inhibitors, GSK-LSD1 and ORY-1001, showed a potent inhibition of the proliferation of MG tumor cells in vitro [43].

On the other hand, the immunohistochemistry and immunofluorescence of the cerebellar tumors in orthotopic Gr3 MB xenografted mice treated with AA7.1 in vivo showed reduced Prune-1 levels and consequently increased the neuronal differentiation markers neuron-specific class III b-tubulin Tuj1 and GFAP [15]. It is possible that disrupting or boosting these functions could affect the ability of cancer to efficiently proliferate. Of interest, the combination of AA7.1 and SP-2577 determines a strong reduction in the protein levels of OTX2 in D425-Med cells. The transcription factor OTX2 controls dorsal mesencephalic neurogenesis. It is required to promote mesencephalic differentiation and suppress cerebellar fate [52]. The loss of OTX2 impairs the identity and fate of progenitors, which undergo a full switch to a coordinated cerebellar differentiation program [52]. Thus, this provides a glimpse into the role that OTX2 could play in the brain differentiation process. We could speculate that OTX2 down-regulation may determine the activation of the pathways of cerebellar differentiation in MB cells. These data have great importance because the use of the combination of AA7.1, as specific targets of Prune-1, and SP-2577 molecules might also result at the interplay between Gr3 and Gr4 MB, with OTX-2 being an important component of Gr4 MB predisposition too [16].

Of interest, our results of the RNA-Seq analysis on primary Gr3 MB show that the combination of AA7.1 and SP-2577 on primary Gr3 MB cells enhances the mRNA levels of the genes implicated in neuronal differentiation, axon genesis, and synapse organization (Appendix A). As presented in Appendix A the RNA-Seq data confirm the up-regulation of the TUBB3 (class III member of the beta tubulin protein family) gene upon both inhibitor treatments of primary Gr3 MB cells, and this gene is a known marker of the neural fate differentiation. Our data showed that the combination of the molecules AA7.1 and SP-2577 might impair the stemness property of Gr3 MB cells to a greater extent than the single molecules, and we showed that this was exerted through the inhibition of the TGF-β regulators Prune-1 and LSD1/KDM1A. We speculate at this time that this would determine a shift to a more neuronal-committed phenotype of tumor cells. Taken all together, the data presented in this paper strengthen the potential use of these two inhibitors in vivo.

We show in this paper that the specific inhibition of Prune-1 and LSD1 using AA7.1 and SP-2577 affects also the phenotype of the components of the TME. Growing evidence has been reported about the positive or negative effects of the immune system on the TME also regarding brain cancers [53,54]. MB is supposed to have a very low level of antitumor immune response due to the low percentage of “effector” T cells, such as granzyme B-expressing CD8+ T cells and natural killer (NK) cells [55]. LSD1/KDM1A was found to be negatively correlated with the expression of CD8+ on T cells and positively correlated with that of PD-L1 [56]. Shen et al. observed also that LSD1 inhibited the response of T cells in the TME of gastric cancer by inducing the accumulation of PD-L1 in exosomes [56]. LSD1/KDM1A inhibition induces the expression of CD8+ T-cell-attracting chemokines in TNBC cell lines [57] and stimulates IFN-dependent antitumor immunity in ovary hypercalcemic-type (SCCOHT) cell lines, promoting PDL1 expression in SCCOHT and ovarian clear cell carcinoma (OCCC) cells [58].

Despite intense research efforts, the mechanisms that reconcile the critical role of epigenetic changes in regulating T-cell exhaustion are now emerging, with critical implications for immuno-oncology. T-cell exhaustion is the impairment in T-cell effector functions as a consequence of chronic antigen exposure and T-cell receptor signaling during chronic infection and cancer [59,60]. Elevated LSD1/KDM1A levels are reported to be a major contributor to the exhausted T-cell phenotype [61]. Targeting LSD1/KDM1A in the exhausted T cells of immunotherapy-resistant mice increased T-cell effector functionality (corresponding to elevated IFN levels and greater T-cell infiltration) [62].

Although the role of LSD1/KDM1A in TGF-β transcriptional regulation is still debated, a recent study from Hong et al. reported that LSD1 promotes non-small-cell lung cancer (NSCLC) metastasis through the TGF-β1/Smad pathway [63]. TGF-β is a master regulator of the expression of immune genes and exerts immunosuppressive regulation in cancers. Both tumor cells and surrounding cells increase TGF-β production, allowing for not only enhanced tumor progression but also the suppression of immune surveillance [64]. In glioma patients, TGF-β suppress immune responses by several mechanisms, including the blockade of the major histocompatibility complex (MHC) class II expression in glioma cells [65,66,67,68,69] and deactivation of natural killer (NK) cells [70] and cytotoxic T cells [71]. T cell adhesion is also reduced by TGF-β, thereby preventing tumor infiltration of lymphocytes into the brain [72,73] while infiltrating lymphocytes are inactivated and induced to undergo apoptosis at the tumor site by TGF-β [71].

As already mentioned, Prune-1 enhances TGF-β cascade and previous studies have shown in in vivo models how Prune-1 influences the TAM polarization and recruitment in TME of TNBC [15,28]. Our finding suggests that the combination of the immunomodulatory molecule AA7.1 with SP-2577 might be crucial to inhibit TGF-β. This action is exerted through the inhibition of TGF-β regulators Prune-1 and LSD1/KDM1A. This is supported by multiple lines of evidence and may determine a shift of immune components of TME from a suppressive phenotype to a more responsive one. In our experiments, we observed the up-regulation of genes that might be responsible for the activation of immune system (Appendix A). In particular, we found high expression of CXCL8 chemokine, also known as interleukin 8 (IL8) (fold 54.19723-fold; *p* = 1.07 e−10) (Appendix A). The CXCL8 responsiveness of T lymphocytes is known to define a CD8+ T-cell subset enriched in perforin, granzyme B, and interferon-γ (IFN γ) and with a high cytotoxic potential [70]. Although the role of CXCL8 is reported to be controversial in cancer, Li et al. [71] found that CXCL8 may be functional as an antitumor immune response in the TME of patients affected by colorectal cancer (CRC) through the stimulation of dendritic cells (DCs), essential for T-cell activation [71]. Thus, we might speculate that the control of cytokine and chemokine “secretomes” is crucial in the TME.

Besides the production and secretion of immuno-suppressive cytokines, other mechanisms could be responsible for cancer immune escape. Among these, impairments in the antigen presentation machinery may play a crucial role [72]. Cancer cells can evade the immune system by altering their ability to process and present antigens [72]. MHC molecules act as a bridge between cancer cell proteins and the immune cells responsible for cytotoxic CD8+ T cells that recognize and attack cancer cells. Some cancer cells can down-regulate the expression of MHC molecules, making them invisible to the immune system [73,74,75]. For example, mutations in the genes encoding the proteasomes or TAP can reduce the ability to generate peptides that can be presented on MHC molecules [76]. The cancer cells’ manipulation of the epigenetic landscape has been shown to play a critical role in suppressing the immune response against cancer and might be important to improve the efficacy of cancer immunotherapy [77]. HDAC inhibitors have been shown to enhance the maturation of dendritic cells and increase the presentation of tumor antigens, thus enhancing the immune response against cancer [78]. Of note, the treatment of Gr3 MB cells with the combination of AA7.1 and SP-2577 resulted in the up-regulation of the genes involved in antigen processing and the presentation of endogenous peptide antigens via MHC class I via the ER pathway, TAP independence, and antigen processing and the presentation of peptide antigens via MHC class Ib. Interestingly, we observed the up-regulation of the RNA level of genes coding for class I histocompatibility antigens—HLA-A (4.01315-fold; *p* = 1.18 e−14) and HLA-G (8.415614-fold; *p* = 2.38 e−27)—and for class II histocompatibility antigens—HLA-DPA1 (3.685418-fold; *p* = 8.93 e−06) and HLA-DPB1 (3.968975-fold; *p* = 7.18 e−07) (Appendix A). Thus, the combined usage of an epigenetic modulator targeting the mechanisms that cancer cells use to evade the immune system might generate a more efficient therapy. Future experiments on regulating T-cell exhaustion using both LSD1/KDM1A and Prune-1 inhibitors need to be performed in vivo to confirm this hypothesis. As described above, these results suggest the use of the combination of these small molecules to enhance in vivo CD8+ T-cell expression, thus enhancing the possibility of having a responsive “hot” TME mediating T-cell-APC responsive action and then using, in combination, any PDL-1 immune check-point inhibitor [79].

Of importance, we found by RNA-Seq analysis an impairment in the mitochondrial metabolism- and energy production-related pathways in primary MB cells treated with the combination of AA7.1 and SP-2577 (Appendix A). Otto Warburg observed that cancers acquire the unusual property of taking up and fermenting glucose to lactate in the presence of oxygen (aerobic glycolysis or Warburg effect), which led him to propose the mitochondrial respiration defects are the underlying basis for aerobic glycolysis and cancer [80]. Not all tumors, however, share this property of aerobic glycolysis. It is also now clear that mitochondrial respiration defects are not generally the cause of aerobic glycolysis, nor are they generally selected during tumor evolution [81]. Several cancer cells, especially cancer stem cells with a high metastatic and tumorigenesis potential, depend more on OXPHOS than the bulk [82]. In several cancers, oncogenic driver mutations promote glycolysis and not mutations that inactivate mitochondrial respiration complexes [83]. Many cancers have increased a mitochondrial DNA (mtDNA) content with relatively higher OXPHOS activity than normal tissues [84]. Depleting mitochondrial DNA from tumor cells by poisoning mitochondrial DNA replication compromises tumorigenesis [85]. The inactivation of the mitochondrial transcription factor TFAM that depletes mitochondria from tumor cells impairs the K-ras lung tumor [86]. The inhibition of OXPHOS through the knockdown of the mitochondrial complex I or V in Drosophila brain tumors caused a decrease in tumor growth and heterogeneity [87]. Regarding brain cancers, Kuramoto et al. [88] found higher expression levels of the mitochondrial transcription factor TFAM in undifferentiated glioma stem cells (GSCs) compared to more differentiated glioma cells expressing the GFAP [88]. Interestingly, the components of the respiratory complexes III and IV were markedly reduced after cell differentiation [88].

Of importance, evidence in the literature suggests that metastatic cells in the brain up-regulate OXPHOS-related genes [89]. According to this, Martirosian et al. [90] observed that MB cells cultured in artificial cerebrospinal fluid (aCSF) displayed high levels of OXPHOS [90]. As Gr3 MB has the highest metastatic potential among all MB, we are convinced that Gr3 MB cells display a high rate of OXPHOS, allowing them to initiate dissemination and metastasis. Small-molecule inhibitors of OXPHOS are currently being developed and tested in clinical trials to treat various forms of cancer [91,92].

Cao et al. [93] reported that LSD1-modulated histone methylation epigenetically regulates nuclear-encoded mitochondrial genes [93]. Experiments of genome-wide binding and transcriptome analyses demonstrated that LSD1 directly stimulated the expression of genes involved in OXPHOS in cooperation with the nuclear respiratory factor 1 (Nrf1) [94]. Thus, this indicates that LSD1 is a key regulator of OXPHOS and mitochondrial metabolism. Also, Prune-1 may have an implication in mitochondrial functions. Human Prune-1 is evolutionarily close to eukaryotic PPXs [17]. The specific reduction in mitochondrial polyP, by S. cervevisiae polyphosphatase scPPX1 expression, significantly modulates mitochondrial bioenergetics, as indicated by the reduction in inner membrane potential and increased NADH levels [95].

Regarding MB, recently, Li et al. [96] identified significantly expressed OXPHOS complex-associated proteins of mitochondria [96]. In particular, in Gr3 MB, the gene COX6B2 (encoding for a subunit of cytochrome c oxidase) was strongly up regulated [96]. Gr3 MB tumors are more resistant to chemotherapy [97]. Linke et al. [98] observed that a Gr3 MB tumor model was characterized by multiple subpopulations with greatly enhanced OXPHOS and tricarboxylic acid (TCA) cycle activity with very high levels of fumarate [98]. While vincristine alone was not sufficient to decrease the cell viability of Gr3 MB tumors, the combination with the NRF2—a fumarate-mediated oxidative stress pathway member—inhibitor significantly enhanced the chemotherapy effect [98]. Phenformin, a mitochondrial complex 1 inhibitor, induced significant cell death in Gr3 MB cells [99]. Then, OXPHOS suppression by using complex-I inhibitors (Phenformin, Rotenone, and IACS-010759), impaired the cell number of HD-MB03 (MYC-amplified Gr3 MB cells) after 24 h of treatment [100]. In particular, IACS-010759 treatment promoted differentiation and suppressed the growth and stemness of Gr3 MB cells [100]. Of note, the oral administration of IACS-010759 impaired tumor growth and prolonged survival in a pre-clinical orthotopic Gr3 MB xenograft model [100]. β-blockers were reported to be able to disrupt mitochondrial bioenergetics, increasing radiotherapy efficacy in MB [101]. Thus, we could hypothesize at this time that OXPHOS plays an important role in MB and its impairment might be crucial. Interestingly, in primary Gr3 MB cells treated with the combination of AA7.1 and SP-2577, we observed a significative reduction in the expression of genes encoding for the subunits of complexes of the mitochondrial electron chain transport and ATP synthase (Appendix A). In more detail, we found the downregulation of the genes of the complex I NADH dehydrogenase (Appendix A). We found a lower expression in the genes of the complex II succinate dehydrogenase SDHAF4 and of the complex IV cytochrome c oxidase COA6 (Appendix A). Of importance, we also observed the down-regulation of the genes that encode for the subunits of ATP synthase ATP5F1A (Appendix A). This evidence allows us to hypothesize the importance of the inhibition of both Prune-1 and LSD1 with our drugs AA7.1 and SP-2577 to impair the mitochondrial energy production of MB cells.

Hence, the combination of WES analyses and gene expression data allowed us to identify a candidate target gene that was mutated with non-synonymous SNV in our Gr3 MB sample and down-regulated with the combination of AA7.1 and SP-2577. The gene was ACSF2 (NM_001288971), mutated in exon4, c.C290A, and p.P97H. This gene encodes for a mitochondrial enzyme member of the acyl-CoA synthetases family that catalyzes the initial reaction in fatty acid metabolism, by forming a thioester with CoA. We reasoned that this protein helps to sustain mitochondrial metabolism, and probably, the aberrant expression of its mutant form has a more pathogenic potential (potential pathogenic variant, Appendix A). The β-oxidation of fatty acids is a way for cancer cells to produce energy and metabolites [102], including the acyl-CoA that enters the mitochondria and participates in the TCA [103] enabling ATP production through OXPHOS [104]. We hypothesize that this enzyme helps to sustain mitochondrial metabolism, and probably, its aberrant expression has pathogenic potential. Interestingly, the gene ACSF2 has been found by others researchers as a biomarker in several different malignancies [105,106]. Of note, by analyzing the dataset Pediatric Brain Cancer (CPTAC/CHOP, Cell 2020, https://www.cbioportal.org/, accessed on 13 February 2024), we observed that, for the gene ACSF2, it is reported the gain of copy-numbers in most MB samples taken into consideration (Appendix A). Thus, this further suggests that the mutated gene identified in this paper may alter the metabolic pathways to produce fuel for cell growth and aggressiveness. This, indeed, strengthens the notion that ACSF2 could be a new potential target for MB therapy, due to its role in energy metabolism. The treatment of primary Gr3 MB cells with the combination of AA7.1 and SP-2577 resulted in a lower RNA expression of ACSF2 compared to the vehicle cells (−3.388010-fold; *p* = 2.19261 e−11) (Appendix A). Thus, this demonstrates the importance of having an approach that allows to overlap genomic and transcriptional signatures and to understand how they influence each other.

In summary, the present study highlights the potential function of the combination of AA7.1 and the epigenetic drug SP-2577 on the blockage of the metastatic axis driven by Prune-1 in Gr3 MB, which is stronger with the combination of the two drugs compared to single treatment with AA7.1. This metastatic axis is responsible for initiation of a pro-migratory phenotype of Gr3 MB tumors, which can then result in metastatic dissemination [15]. Since LSD1/KDM1A is a putative positive regulator of Prune-1 in Gr3 MB, the inhibition of Prune-1 by AA7.1 is reinforced by the inhibition of LSD1/KDM1A. In addition, our data demonstrated that the combination of AA7.1 and SP-2577 positively affected the neuronal commitment and might be responsible for the activation of the cytotoxic components of the immune system of the TME. A model describing the action of the combination of the two drugs is presented (Figure 5C). The overexpression of LSD1 in Gr3 MB leads to the activation of PRUNE-1 transcription and the activation of the TGF-β pathway through SMAD2/3 phosphorylation and nuclear translocation (PTEN) inhibition. Upon LSD1/KDM1A inhibition by SP-2577 in combination with the PRUNE-1 inhibitor AA7.1, LSD1 is not able to enhance PRUNE1 transcription and this, in turn, results in the inhibition of the activation of the TGF-β pathway. Then, PTEN is found up-regulated. Overall, these phenomena determine the higher neuronal commitment of tumor cells and, in particular, T cells then turn into a cytotoxic phenotype. The gene expression data further confirmed the significant impairment in mitochondrial metabolism and OXPHOS. Taken all together, we also observed the capability to block the mitochondrial electron chain transport and consequently the oxidative phosphorylation, depriving tumor cells of metabolic energy produced in the mitochondrion. Of interest, recently, Zeuner et al. [107] reported the identification of the combination of BCL-XL inhibitors with epigenetic inhibitors of class I HDAC as a potential new approach for the treatment of MYC-amplified Gr3 MB cells [107]. Thus, this demonstrates the importance of therapies based on the combined usage of epigenetic drugs to modulate MB tumors. In conclusion, our data allow us to highlight the efficacy of the combination of a small-molecule anti-Prune-1 inhibitor (AA7.1) and an epigenetic drug targeting LSD1/KDM1A (SP-2577) and the identification of a new potential approach for the therapeutic treatment of metastatic MB.

However, a more comprehensive characterization of the anti-Prune-1 molecule AA7.1 and LSD1 inhibitor SP-2577, followed by an in vivo validation of our findings using a mouse model of Gr3 MB, is still required. We believe that an integrated multidimensional approach involving genomic mutational signatures and transcriptomic features could be helpful to identify potential actional targets and stratify the patients for treatments. An integrated approach is important to understand the different responses to the therapies. In conclusion, we would like to propose the combination of the two compounds (AA7.1 and SP-2577) for the treatment of patients affected by Gr3 MB, for which efficient therapies are current lacking. All in all, considering our preliminary data of the efficacy of the treatment to target SHH and Gr4 MB, future efforts could also address other MB aggressive subgroup therapies.

## 4. Materials and Methods

### 4.1. Data Source

The expression data of medulloblastoma patients were collected using publicly available datasets on R2: Genomics Analysis and Visualization Platform (https://hgserver1.amc.nl/cgi-bin/r2/main.cgi accessed on 20 February 2024). ChIP-Seq data were obtained from the UCSC Genome Browser (https://genome.ucsc.edu/ accessed on 20 February 2024).

### 4.2. Cell Culture

D425-Med cells and D283-Med cells were grown in a humidified 37 °C incubator with 5% CO2. The cells were cultured under feeder-free conditions using Dulbecco’s modified Eagle’s medium (DMEM; 41966-029; Gibco, New York, NY, USA) with 10% fetal bovine serum (10270-106; Gibco), 2 mM L-glutamine (25030-024; Gibco), and 1% penicillin/streptomycin (P0781; Sigma-Aldrich, St. Louis, MI, USA), with the medium changed daily. The cells were dissociated with a Trypsin-EDTA solution (T4049, Sigma-Aldrich) when the cultures reached ~80% confluency. The primary Gr3 MB cells were generated through the trypsinization of the biopsy derived from the neurosurgical resection of a metastatic Gr3 MB. The primary cells were cultured using Dulbecco’s modified Eagle’s medium (DMEM; 41966-029; Gibco) and minimum essential medium (MEM; 11090081; Gibco) in a ratio of 1:1 with 10% fetal bovine serum 10270-106; Gibco), 2 mM L-glutamine (25030-024; Gibco), and 1% penicillin/streptomycin (P0781; Sigma-Aldrich), with the medium changed daily.

### 4.3. In Vitro Treatments

D425-Med cells (3 × 10^5^) and primary Gr3 MB cells (3 × 10^5^) were placed in 6-well plates and treated with 100 µM AA7.1, 22.5 µM SP-2577, the combination of 100 µM AA7.1 and 22.5 µM SP-2577, or with 0.001% DMSO as the vehicle control. Cellular pellets were collected after 24 h of treatment and used for protein or RNA extraction.

### 4.4. Cell Proliferation Assays (Cell Index)

The real-time cell proliferation analysis for the cell Index (i.e., the cell-sensor impedance was expressed every 2 min as a unit called “Cell Index”) was carried out by using an xCELLigence RTCA Analyzer. The D425-Med cells (1.5 × 10^4^) and D283-Med cells (1.5 × 10^4^) were plated an in xCELLigence E- plate 16; #05665817001; Acea Biosciences. After 2 h, the cells were treated with the indicated concentrations of AA7.1, TCP, and SP-2577; the vehicle-treated cells were the negative control. The impedance was measured every 2 min over 24 h. The IC_50_ values were calculated through a nonlinear regression analysis performed with Graph Pad Prism 9 ([inhibitor] vs. response (three parameters)). Data represent the means ± SD of three independent experiments.

### 4.5. Immunoblotting

The cells were lysed in 20 mM sodium phosphate, pH 7.4, 150 mM NaCl, 10% (*v*/*v*) glycerol, 1% (*w*/*v*) sodium deoxycholate, and 1% (*v*/*v*) Triton X-100, supplemented with protease inhibitors (Roche). The cell lysates were cleared by centrifugation at 16,200× *g* for 30 min at room temperature, and the supernatants were removed and assayed for protein concentrations with a Protein Assay Dye Reagent (BioRad). The cell lysates (20 µg) were resolved on 10% SDS-PAGE gels. The proteins were transferred to PVDF membranes (Millipore). After 1 h in a blocking solution with 5% (*w*/*v*) dry milk fat in Tris-buffered saline containing 0.02% (*v*/*v*) Tween-20, the PVDF membranes were incubated with the primary antibody overnight at 4 °C: anti-β-ACTIN (1:10,000; A5441; Sigma); anti-PRUNE-1 (1:1000; ab88613; Abcam); anti-OTX2 (1:250; ab130238; Abcam); anti-SMAD2-phospho S467 (1:250; ab53100; Abcam); anti-SMAD3-phospho S423-425 (1:250; 9520; Cell Signaling, Danvers, MA, USA); and anti-PTEN (1:500; 9559; Cell Signaling). The membranes were then incubated with the required secondary antibodies for 1 h at room temperature, as secondary mouse or rabbit horseradish-peroxidase-conjugated antibodies (NC 15 27606; ImmunoReagents, Inc.), diluted in 5% (*w*/*v*) milk in TBS-Tween. The protein bands were visualized by chemiluminescence detection (Pierce-Thermo Fisher Scientific Inc., Rockford, IL, USA). The densitometry analysis was performed with the ImageJ software. The peak areas of the bands were measured on the densitometry plots, and the relative proportions (%) were calculated. Then, the density areas of the peaks were normalized with those of the loading controls, and the ratios for the corresponding controls are presented as fold-changes.

### 4.6. Brain MRI

MRI studies were acquired at 1.5 T (Achieva, Philips Medical Systems, Best, The Netherlands). Inclusion criteria were the availability of a T2w and a T1w sequence (axial planes covering the whole ICV, with the sequence parameters selected according to the clinical needs), without major artifacts related to patient movements or alterations of the main magnetic field homogeneity due to the presence of ferromagnetic devices. The analyzed T2w images were acquired using TurboSpin-Echo sequences. T1-weighted images were obtained by 3D TurboField-Echo.

### 4.7. Immunohistochemistry

For the immunohistochemical studies, tumor tissues fixed in formalin and included in paraffin were used. Paraffin sections (thickness, 3 μm) of the tumor specimens were deparaffinized in Bioclear (06-1782D; Bio-Optica, Milan, Italy) for 30 min, rehydrated in 100%, 90% and then 70% ethanol, and washed with PBS and then PBS containing 0.02% Triton-X 100 (215680010; Acros Organics, Geel, Belgium). The streptavidin–biotin–peroxidase method was used for the detection of synaptophysin (1:200; GA660, Dako, Glostrup, Denmark). The antigenic was recuperated with sodium citrate (pH 6.0) for 30 min. The endogenous peroxidase activity was blocked with 3% hydrogen peroxidase (H2O2). Incubation with primary antibodies was performed overnight at 4 °C. Incubation with secondary antibodies was performed for 30 min at room temperature. Then, antigen–antibody binding was visualized.

### 4.8. Immunofluorescence

D425-Med cells were plated and grown (1×103 cells) on coverslips. After 24 h of treatment, the cells were fixed in 4% paraformaldehyde, permeabilized for 10 min with 0.1% Triton X-100 (215680010, Acros Organics) diluted in PBS. The cells were then washed with PBS and blocked with 3% BSA (A9418, Sigma-Aldrich) in PBS for 1 h at room temperature. The samples were incubated with the appropriate primary antibodies overnight at 4 °C: anti-glial fibrillary acidic protein (GFAP; z0334, Dako, 1:100). After washing twice with PBS, the samples were incubated with the appropriate secondary antibody at room temperature for 1 hour: anti-rabbit Alexa Fluor 647 (#A-21245; ThermoFisher) was used as the secondary antibody. DNA was stained with 4′,6-diamidino-2-phenylindole (1:1000; #62254, Thermo Fisher Scientific). Confocal microscopy was carried out using a laser scanning confocal microscope (LSM 510 META, Zeiss), with the 63× oil immersion objective. Quantification of the fluorescence intensity was performed with ImageJ software (version 1.52r; https://imagej.nih.gov/ij/index.html, accessed on 20 February 2024), and the corrected total cell fluorescence (CTCF) was estimated according to the following equation: CTCF= integrated density− (area of selected cell × mean background fluorescence). Mantra Quantitative Pathology Workstation was used to extract images. 

### 4.9. Genomic DNA Extraction

The genomic DNA from the blood/biopsy tissue was extracted using the All-prep DNA/RNA mini kit (50) (lot: #172027819, ref: #80204). A mutation analysis was performed with the next-genome sequencing (NGS) with whole-exome sequencing (WES) approach.

### 4.10. Whole-Exome Sequencing

The DNA was amplified at the level of all genomic exons. The assembly of the library, the sequencing, and the raw analysis of the data were carried out at the company Macrogen (Quality Management System registration certificate—ISO9001:2015 N.: M 89418). The analysis of the sequences was carried out using the integrated primary analysis software called RTA (Real Time Analysis). Base call files expressed in binary code were converted to FASTQ by the Illumina bcl2fastq v2.20.0 package. Minimum Coverage Achieved: 99×.

### 4.11. Analysis of Variants

The variants were analyzed according to the American College of Medical and Genomics guidelines (“Standards and guidelines for the interpretation of sequence variants: a joint consensus recommendation of the American College of Medical Genetics and Genomics and the Association for Molecular Pathology” [108] and somatic variants were analyzed taking into account the frequency of mutations in public databases (GnomAD) and data on pathogenicity and variant conservation present in the public databases REVEL (rare exome variant ensemble learner), GERP (Genomic Evolutionary Rate Profiling), CADD (Combined Annotation Dependent Depletion), MUTscore {Quinodoz, 2022 #114}, and InterVar {Li, 2017 #113}. In more detail, the variants were filtered and prioritized using the following: for the row “gnomAD_genome_ALL”, we considered <0.01 values; for the row “cadd16”, we considered > o = 15 values; for the row “Gerp++”, we filtered for >2 or =2 values; for the row “BG (Intervar)”, we filtered for “pathogenic, likely pathogenic and uncertain significance”; and for the row “AE (REVEL)”, we filtered for >0.5 or =0.5 values. Mutscore of 0.12 is considered the threshold score value below which the variants were predicted to be likely benign/benign. The detected genetic variants were identified and classified using standard nomenclature (HGVS and HUGO Gene Nomenclature Committee—https://www.genenames.org, accessed on 20 February 2024).

### 4.12. RNA Extraction and qPCR Assays

The RNA samples were extracted using TRIzol RNA Isolation Reagent (15596026, Invitrogen), according to the manufacturer’s instructions. Reverse transcription was performed with 5× All-In-One RT MasterMix (G592; ABM, New York, NY, USA), according to the manufacturer’s instructions. The reverse transcription products (cDNA) were amplified by qRT-PCR using an RT-PCR system (2700; Applied Biosystems, Foster City, CA, USA). The cDNA preparation was through the cycling method, by incubating the complete reaction mix as follows: cDNA reactions: (37 °C for 10 min and 60 °C for 15 min); heat inactivation: 95 °C for 3 min; and hold stage: 4 °C. The targets were detected with the SYBR green approach, using BrightGreen BlasTaq 2× PCR MasterMix (G895; ABM). Human ACTB was used as the housekeeping gene to normalize the quantification cycle (Cq) values of the other genes. These runs were performed on a PCR machine (Quantstudio5, Life Technologies) with the following thermal protocol: hold stage: 50 °C for 2 min; denaturation step: 95 °C for 10 min; denaturation and annealing (×45 cycles): 95 °C for 15 s and 60 °C for 60 s; and melt curve stage: 95 °C for 15 s, 60 °C for 1 min, and 95 °C for 15 s. The relative expression of the target genes was determined using the 2^−ΔΔCq^ method, as the fold increase compared to the controls. The data are presented as the means ± SD of the 2^−ΔΔCq^ values (normalized to human ACTB) of three replicates.

The primer sequences for the qPCR analyses included the following:

ACTB Forward: GACCCAGATCATGTTTGAGACCTT.

ACTB Reverse: CCAGAGGCGTACAGGGATAGC.

PRUNE-1 Forward: TTCGGGATGAGATTGACCTCC.

PRUNE-1 Reverse: GCTCGATGGGTCGATGGTCT.

OTX2 Forward: CGAGAGGAGGTGGCACTGAA.

OTX2 Reverse: GCGGCACTTAGCTCTTCGATT.

### 4.13. RNA Sequencing (RNA-Seq)

For the RNA isolation and library construction and sequencing, the total RNA was isolated from primary medulloblastoma cells using the TRIzol RNA Isolation Reagent (#15596018; Ambion, Thermo Fisher Scientific). It was then quantified in a NanoDrop One/OneC Microvolume UV-Vis spectrophotometer (Thermo Scientific), checked for purity and integrity, and submitted to Macrogen Europe B.V. for sequencing. Libraries were prepared using TruSeq Stranded mRNA Library Prep kits, according to the protocols recommended by the manufacturer (i.e., TruSeq Stranded mRNA Reference Guide # 1000000040498 v00). The trimmed reads were mapped to the reference genome with HISAT2 (https://ccb.jhu.edu/software/hisat2/index.shtml, accessed on 20 February 2024), a splice-aware aligner. After the read mapping, Stringtie (https://ccb.jhu.edu/software/stringtie/, accessed on 20 February 2024) was used for transcript assembly. The expression profile was calculated for each sample and transcript/gene as read counts, FPKM (fragment per kilobase of transcript per million mapped reads), and TPM (transcripts per kilobase million).

### 4.14. Clustering of Primary Gr3 MB Cells

The sub-classification of our primary tumor was made by extracting RNA from biopsy using the TRIzol protocol. The MB molecular subtype was assigned by using the MedulloClassifier package, which is reported to predict amongst the 4 subtypes of MB—WNT, SHH, Gr3, and Gr4—with a median accuracy of 97.8% (https://github.com/d3b-center/medullo-classifier-package, accessed on 20 February 2024). The prediction was carried out using as input the RNAseq expression data in the form of log2-transformed FPKM. The analyzed sample was predicted as belonging to Gr3 MB with a p-value of 0.005.

### 4.15. Analysis of Differentially Expressed Genes

The differentially expressed genes analysis was performed on a comparison pair (treated vs untreated cells). The read count value of the known genes obtained through the -e option of the StringTie was used as the original raw data. During data preprocessing, low quality transcripts were filtered out. Afterwards, a trimmed mean of M-values (TMM) normalization was performed. The statistical analysis was performed using Fold Change, with exactTest using edgeR per comparison pair. For significant lists, a hierarchical clustering analysis (Euclidean method, complete linkage) was performed to group the similar samples and genes. These results were depicted graphically using a heatmap and dendrogram. For the enrichment test, which is based on Gene Ontology (http://geneontology.org/, accessed on 20 February 2024), DB was carried out with a significant gene list using the g:Profiler tool (https://biit.cs.ut.ee/gprofiler/ accessed on 20 February 2024). The pathway enrichment analysis was performed using GSEA (https://www.gsea-msigdb.org/gsea/index.jsp accessed on 20 February 2024). The network of enriched pathways were built by using Cytoscape (https://cytoscape.org/, accessed on 20 February 2024).

### 4.16. Statistical Analysis

The measurement data (with normal distribution) are presented as the mean ± standard deviation. The data representative of two or three independent experiments were analyzed using paired two-tailed *t*-tests (Student’s *t*-tests). A value of *p* < 0.05 was regarded as a statistically significant difference. The correlation analysis was carried out through R2 (Genomics Analysis and Visualization Platform; http://r2.amc.nl, accessed on 20 February 2024). The protein densitometry for Western blotting quantification was carried out using the ImageJ software to calculate the relative and normalized densities of the peaks corresponding to the bands for the proteins of interest and those relative to the loading control bands.

## 5. Patents

An European Patent resulting from the work reported in this manuscript was presented. PCT/EP2023/079960, with previous Deposit of the 28-10-2022 (EP22204452.1) and a second Deposit 26-10-2023. Title “PRUNE_1 inhibitors and therapeutic use there of”. 

## Figures and Tables

**Figure 1 ijms-25-03917-f001:**
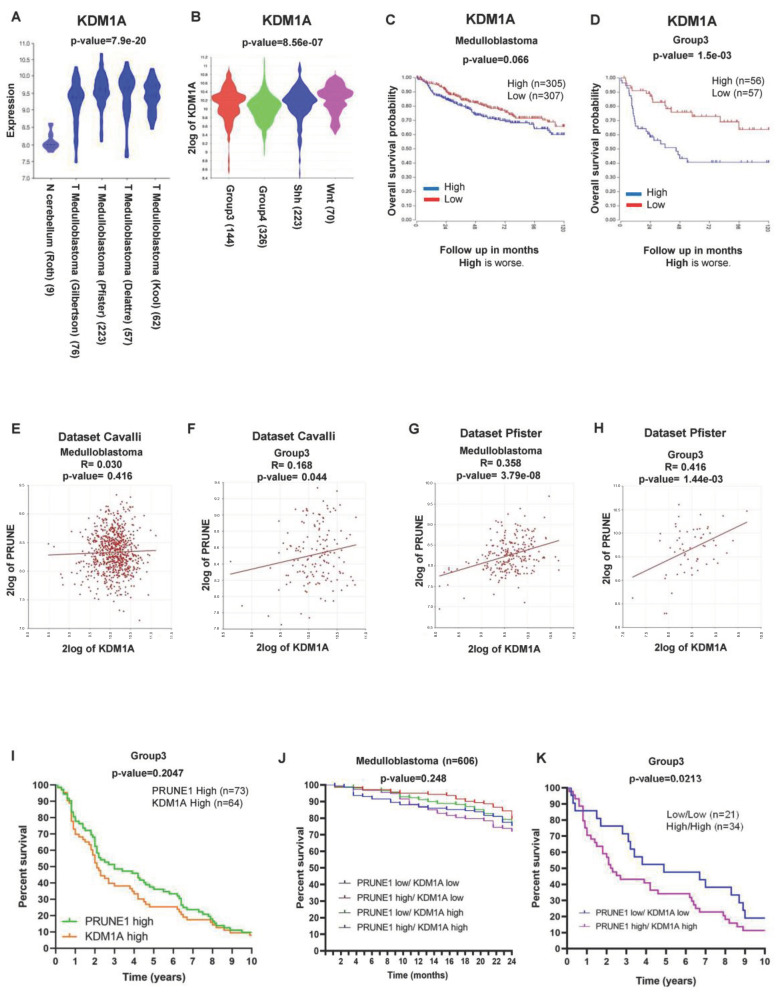
(**A**) Expression of LSD1/KDM1A derived from multiple datasets of MB (Kool, *n* = 62; Delattre, *n* = 57; Pfister, n = 223; Gilbertson, *n* = 76) compared to that in normal cerebellum (Roth, *n* = 9) (*p* < 0.001). LSD1/KDM1A was highly expressed in MB. (**B**) RNA log2 expression of LSD1/KDM1A derived from publicly available dataset of MB (Cavalli; *n* = 763; *p* < 0.001), grouped according to the molecular group disease variants. LSD1/KDM1A was highly expressed in Gr3 MB. (**C**) Kaplan-Meyer analysis for event-free survival (EFS) of MB and its groups patients according to LSD1/KDM1A expression levels. Patients who showed higher levels of LSD1/KDM1A expression (*n* = 305) showed shorter event-free survival compared to those with lower levels of LSD1/KDM1A expression (*n* = 307) (*p* = ns). Red = Low; Blue = High. (**D**) Gr3 MB patients who showed higher levels of LSD1/KDM1A expression (*n* = 56) showed significantly shorter event-free survival compared to those with lower levels of LSD1/KDM1A expression (*n* = 57) (*p* < 0.001). (**E**) Correlation between the expression of LSD1/KDM1A and PRUNE1 in MB (Cavalli; *n* = 763; R = 0.030; *p* = ns). (**F**) Correlation between the expression of LSD1/KDM1A and PRUNE1 in MB (Cavalli; *n* = 144; R = 0.168; *p* < 0.001). (**G**) Correlation between the expression of LSD1/KDM1A and PRUNE1 in MB and (Pfister; *n* = 223; R = 0.358; *p* < 0.001). (**H**) Correlation between the expression of LSD1/KDM1A and PRUNE1 in Gr3 MB (Pfister; *n* = 56; R = 0.416; *p* < 0.001). (**I**) Survival curves of patients affected by Gr3 MB showing PRUNE1 high (*n* = 73) or LSD1/KDM1A high (*n* = 64; *p* = ns). Green = PRUNE1 High; Orange. (**J**) Combined survival data show that a high expression of both PRUNE1 and LSD1/KDM1A is correlated with a lower percentage of survival in MB (*n* = 606; *p* = ns). Blue = PRUNE1 Low/KDM1A Low; Red = PRUNE1 High/KDM1A Low; Green = PRUNE1 Low/KDM1A High; Violet = PRUNE1 High/KDM1A High. (**K**) Combined survival data show that a high expression of both PRUNE1 and LSD1/KDM1A is correlated with a lower percentage of survival Gr3 MB (low/low = 21; high/high = 34) (*p* < 0.05). Blue = PRUNE1 Low/KDM1A Low; Violet = PRUNE1 High/KDM1A High.

**Figure 2 ijms-25-03917-f002:**
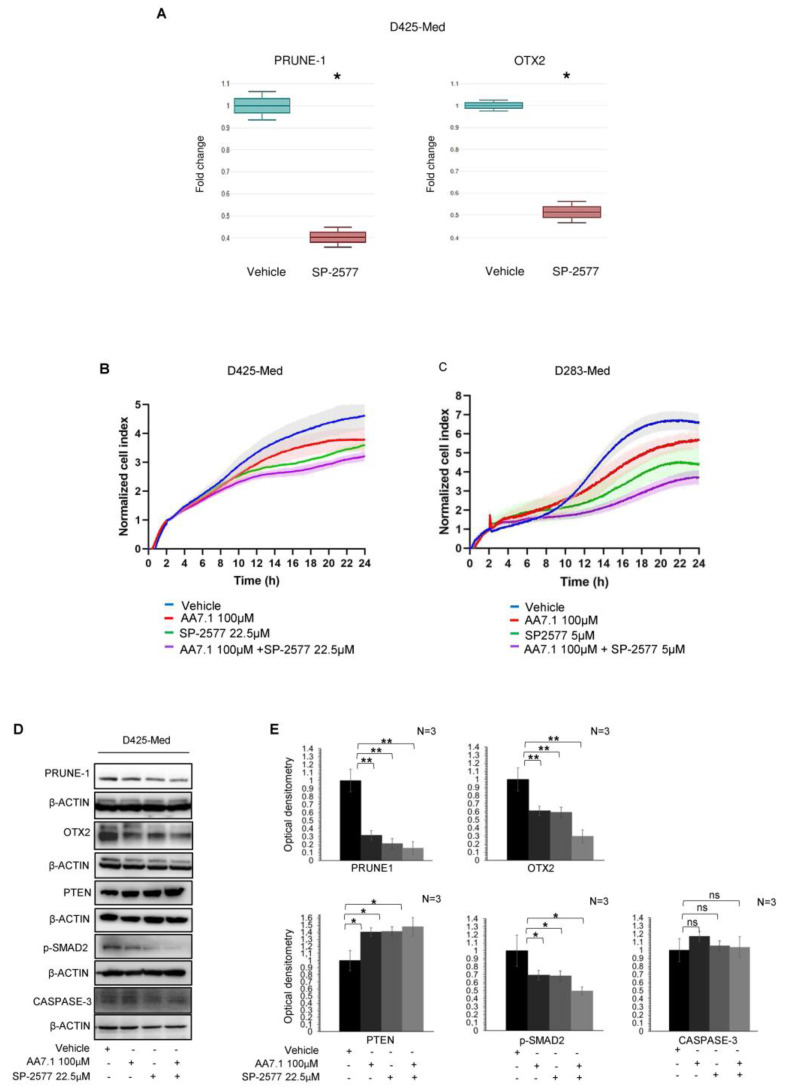
(**A**) Quantification of mRNA abundance relative to untreated control cells (fold change) for the human PRUNE1 and OTX2 genes. RT-PCR analysis of RNA extracted from D425-Med cells treated with SP-2577. Data are means ± SD. * *p* < 0.05, (paired two-tailed student’s *t*-tests; *n* = 2). (**B**) Real-time cell proliferation analyses for the Cell Index. D425-Med cells were plated and treated with 100 μM AA7.1, 22.5 μM SP-2577 and the combination of two; with vehicle-treated cells were the negative control. Impedance was measured every 2 min over 24 h. The graph showing “normalized cell index” was generated using Graph Pad Prism 9. (**C**) Real-time cell proliferation analyses for the Cell Index. D283-Med cells were plated and treated with 100 μM AA7.1, 5 μM SP-2577 and the combination of two; with vehicle-treated cells were the negative control. Impedance was measured every 2 min over 24 h. The graph showing “normalized cell index” was generated using Graph Pad Prism 9. (**D**) Western blotting analysis using antibodies against the indicated proteins on total protein lysates obtained from D425-Med cells treated with 100 μM AA7.1, with 22.5 μM SP-2577 and the combination of two. Vehicle cells were used as negative control. (**E**) Optical densitometry quantification of Western blotting results. Data are means ±SD. * *p* < 0.05, ** *p* < 0.005, ns: no statistical (paired two-tailed student’s *t*-tests; *n* = 3).

**Figure 3 ijms-25-03917-f003:**
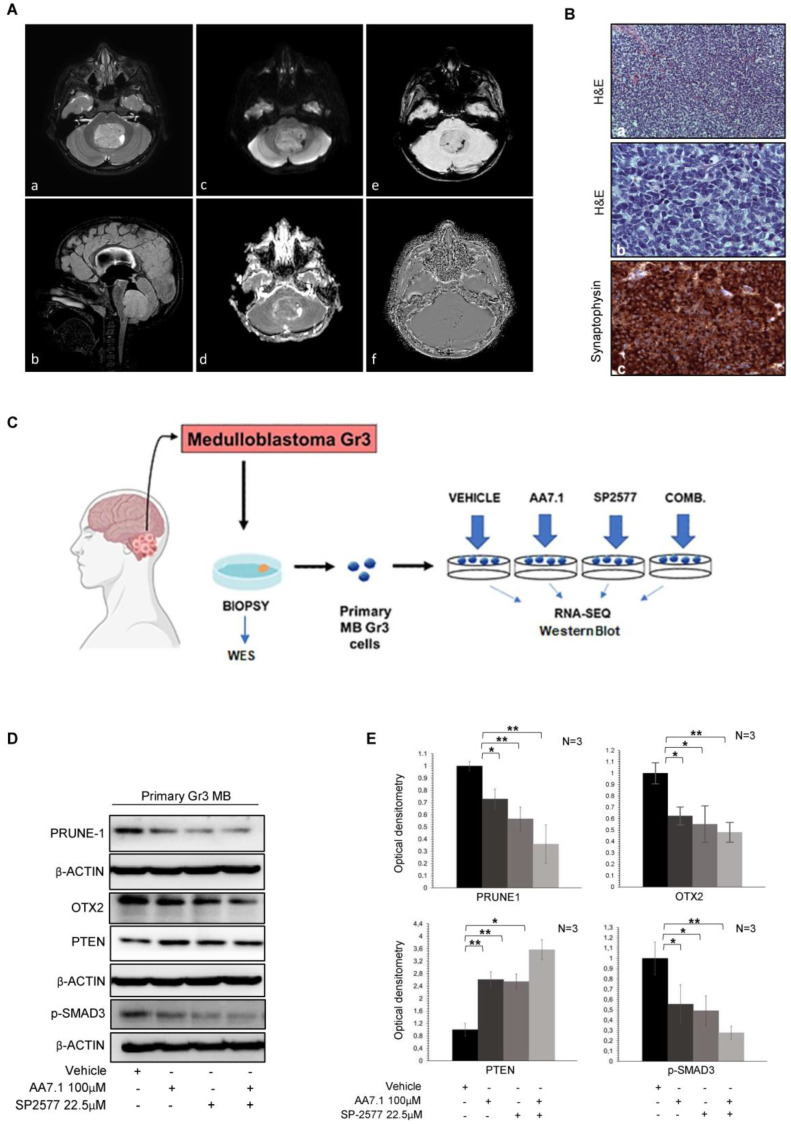
(**A**) Pre-surgery MRI scan of Classical Medulloblastoma in a patient affected. Axial T2-weighted image (**a**) and sagittal 3D-FLAIR reconstruction (**b**), diffusion weighted imaging (**c**) and ADC map (**d**), magnitude (**e**) and filtered phase (**f**) susceptibility weighted imaging. (**B**) (**a**) Classic Medulloblastoma: Typical syncytial arrangement of undifferentiated cells (HE × 200). (**b**) Classic Medulloblastoma: Poorly differentiated embryonal cells with hyperchromatic and variably shaped nuclear (HE × 600). (**c**) Classic Medulloblastoma: Diffuse expression of synaptophysin (×400). (**C**) Schematic representation of the experiments of RNA-Seq on primary Gr3 MB cells. (**D**) Western blotting analysis using antibodies against the indicated proteins on total protein lysates obtained from primary Gr3 MB cells treated with 100 μM AA7.1, with 22.5 μM SP-2577 and the combination of two. Vehicle cells were used as negative control. (**E**) Optical densitometry quantification of Western blotting results. Data are means ± SD. * *p* < 0.05, ** *p* < 0.01 (paired two-tailed student’s *t*-tests; *n* = 3).

**Figure 4 ijms-25-03917-f004:**
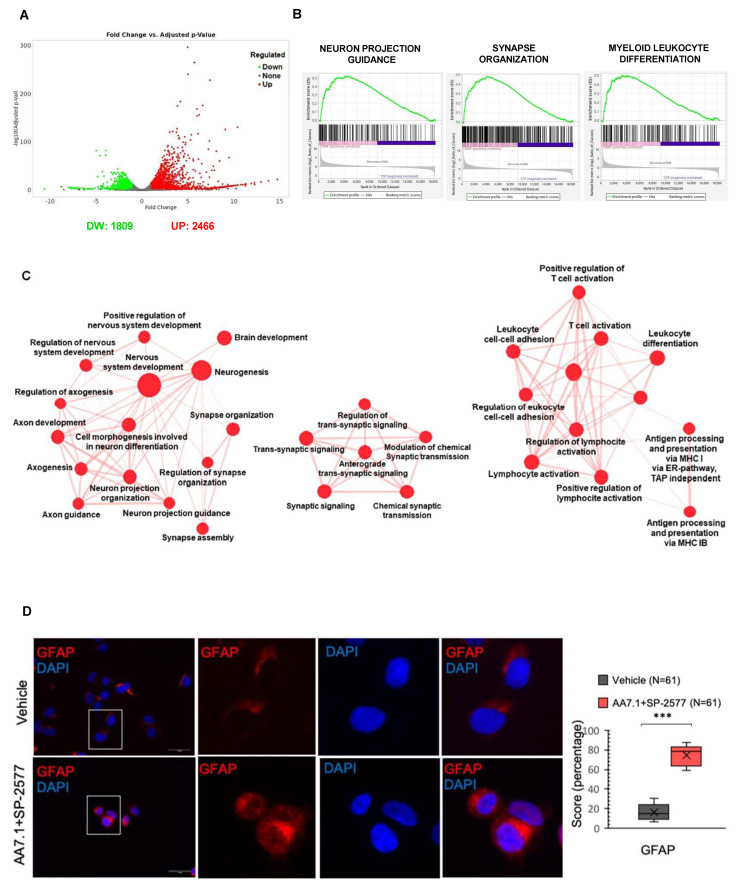
(**A**) Volcano plot of differentially gene expression in Gr3 MB cells treated with 100 μM AA7.1 and 22.5 μM SP-2577 versus vehicle cells. RNA-seq data is deposited with Ebi.ac.uk “BioStudies” portal code “MTAB-13616”. (**B**) GSEA showing that the pathways of neuron projection guidance, synapse organization and myeloid leukocyte differentiation include genes mostly up-regulated in cells treated with the combination of AA7.1 and SP-2577 compared to vehicle cells. (**C**) Cytoscape representation of the network of up-regulated biological functions when the combination of AA7.1 and SP-2577 is used. (**D**) Immunofluorescence staining with an antibodies against GFAP protein in D425-Med cells treated for 24 h with AA7.1 and SP-2577. The graph showing the intensity of fluorescence was shown on the right. Magnification ×63. Scale bar, 5 μm. Blue = DAPI; Red = GFAP. *** *p* < 0.001.

**Figure 5 ijms-25-03917-f005:**
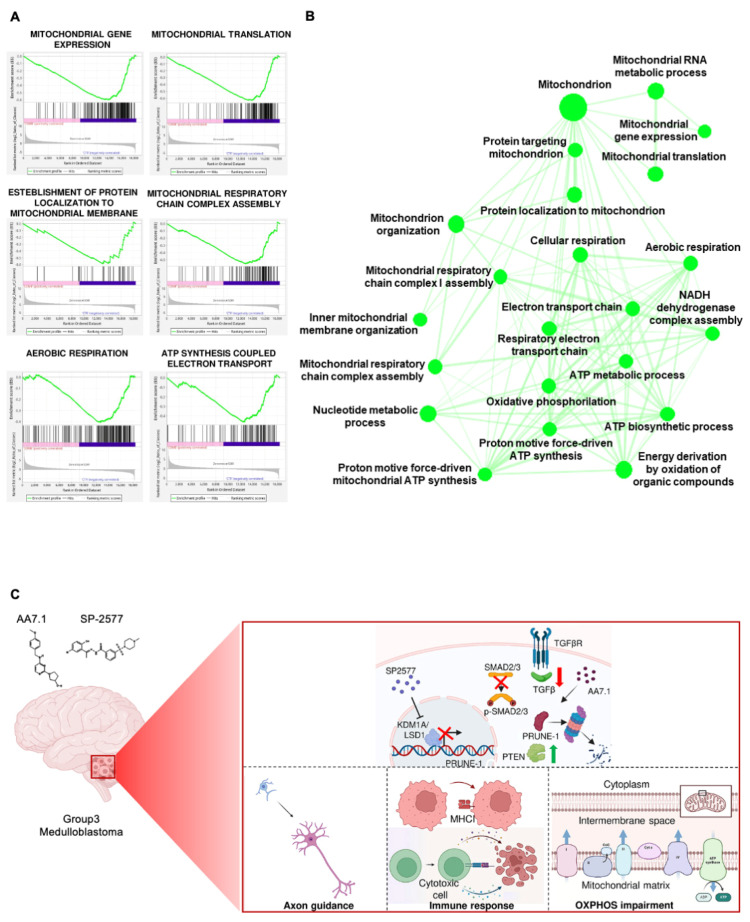
(**A**) GSEA showing that the pathways of mitochondrial gene expression, organization and metabolism include genes mostly down-regulated in cells treated with the combination of AA7.1 and SP-2577 compared to vehicle cells. (**B**) Cytoscape representation of the network of down-regulated biological functions when the combination of AA7.1 and SP-2577 is used. (**C**) Schematic representation of our model. The overexpression of LSD1 in Gr3 MB positively modulates PRUNE-1 expression via binding to its promoter region leading to activation of TGF-β pathway, through SMAD2/3 phosphorylation and nuclear translocation and PTEN inhibition. AA7.1 is responsible for the proteasomal-dependent degradation of Prune-1 protein [15]. When LSD1/KDM1A inhibitor SP-2577 is used in combination with PRUNE-1 inhibitor AA7.1, LSD1 is not able to enhance PRUNE-1 transcription. Then the activation of TGF-β pathway is lower and PTEN is up-regulated. This determines a deregulation of genes involved in neuronal commitment, antigen presenting cells, a cytotoxic mediated T cell response and a significant impairment of mitochondrial metabolism and OXPHOS. Created with Biorender.

**Table 1 ijms-25-03917-t001:** Table of pathogenic variants identified by WES analysis.

Gene Symbol	ID Transcript	Exon	DNA Base Positions	Amino Acid Changes	Variant Function	Gene Description
**DAPK1**	NM_001288729	exon11	c.C938A	p.S313X	stopgain	Death-associated protein kinase 1 is a positive mediator of gamma-interferon-induced programed cell death. It is a tumor suppressor candidate.
**HSPH1**	NM_001286503	exon2	c.G124T	p.G42X	stopgain	This gene encodes a member of the heat shock protein 70 family of proteins. An elevated expression of this protein has been observed in numerous human cancers.
**LRP1B**	NM_018557	exon25	c.G4168T	p.G1390X	stopgain	This gene encodes a member of the low density lipoprotein (LDL) receptor family. The disruption of this gene has been reported in several types of cancer.

## Data Availability

Data supporting reported results Rna-seq Data can be found, at links to publicly archived datasets E-MTAB-13616 (https://www.ebi.ac.uk/biostudies/arrayexpress/studies/E-MTAB-13616?key=d516fa88-3ccb-4583-8d9c-9592701d59e4, accessed on 20 February 2024) analyzed or generated during the study.

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
