# Peer review of "Targeting Group 3 Medulloblastoma by the Anti-PRUNE-1 and Anti-LSD1/KDM1A Epigenetic Molecules"

_ijms, 2024, doi:10.3390/ijms25073917_

Round 1

Reviewer 1 Report

Comments and Suggestions for Authors

LSD1/KDM1A transcriptionally regulates PRUNE-1 which drives MB3 via OTX2-25 TGF-PTEN. Authors look at inhibiting this control axis at PRUNE-1 or LSD1 with inhibitors AA7.1 and SP-2577, respectively.

They conclude, combination of these two small molecules could be used in a second line treatment in advanced therapeutics against Gr3 MB.

Inhibition lead to differentiation and microenvironment cytotoxicity, but also impaired ox phosphorylation through mitochondria.

Abstract

Generally well written. Describes study well.

Note, more commonly inhibition of ox phos is considered oncogenic such that the inhibition of OP here needs at least monitoring.

Introduction

Generally a nice intro to MB and features of each grounds the reader well to the  MB groups in general.

The introduction is probably overlong with a little excess detail around LSD1 action/targeting in particular that could be pared a little and partially move to the discussion although clearly background and stage setting for PRUNE1 and LSD1 needs to occur to some extent.

From the authors earlier paper, it appeared that PRUNE-1 was over-expressed and PTEN consequently reduced in groups 3 and 4 MB accepting that some pathway components were higher in G3 v G4 eg NME-1. Do the authors feel that G4 MB may also benefit with possibly an extension study in the future or do they feel it unlikely  based on previous data.

Results

Figure 1B shows impressive detriment with KDM1A expression in Grp 3 and subtle in MB as a whole. This suggests there may not be a detriment at all with some types, suggest showing curves for all sub-types individually here to clarify the prognostic specificity for Gp3.

The Gp3 curves for high/high v low/low in 1D look less impressive than KDM1A alone in 1B – is PRUNE1 expression adding prognostic data to the model or is KDM1A alone just as good or better (clearly numbers are low).

The PRUNE1-KDM1A link is also underwhelming for all MB but looks solid for Grp 3, suggest show all individual types here also.

The epigenetic KDM1A chip data is in SH-SY5Y  cells – are these MB Gp3 sub-type? If so, was a difference seen in an example of another sub-type considering the lower correlation outside Gp3 MB?

Was the inhibitory activity of LSD1 blockade also tested on other MB types?

The fall in OTX2 appears more prominent than that of PRUNE1 in Fig 2D. Do you feel the former is still due to the latter?

Did you test the impact of the two inhibitors on the primary MB cells or just look at protein expression changes?

The reactivation of differentiating pathways and cytotoxic immune pathways is nice and fits with benefit.

I was a little surprised that mitochondrial gene expression patterns suggested an inhibition of ox phos. It is also slightly concerning as malignant cells tend to channel energy metabolism through glycolysis (ie the Warburg effect as the authors reference) presumptively to repurpose Acetyl CoA into other pathways and use less ox phos with ox phos upreg usually being associated with benefits.

Discussion

When cells differentiate it can impact response to other treatment components eg chemo or RT, possibly with reduced proliferation reducing sensitivity, but also as a counterpoint, loss of EMT and loss of stem cell characteristics can also increase sensitivity. I presume combinations have not yet been done but bears discussion.

Agree that discussion regarding the degree to which this may be a group 3 restricted benefit is warranted as Gp4 appeared to have some similar changes from authors previous paper.

The discussion around modulation of components of the TME is warranted as this could be a significant strength of the therapy bearing in mind MBs low immunogenicity generally.

Regarding the mitochondrial impacts, the authors do reference increased OXPHOS in some cancers cf normal such that a treatment inhibiting OXPHOS could be beneficial but this is not, I feel, where the weight of published literature sits across cancers. There could be an MB-specific effect here as they suggest. Could mention that dissecting out this effect by specific OXPHOS inhibition to make sure this isn’t detrimental could be considered in the future.

The mutation of mitochondrial acyl-CoA ligase in this case is interesting. Raises the question as to whether the mitochondrial effects would be seen if a comparator wild type case was analyzed.

Author Response

We would like to thank the reviewers and the Editor for the constructive comments which helped us to improve our manuscript.

Here below, we provide a point-by-point response to the concerns raised.

Reviewer 1

Comments and Suggestions for Authors

LSD1/KDM1A transcriptionally regulates PRUNE-1 which drives MB3 via OTX2-25 TGF-PTEN. Authors look at inhibiting this control axis at PRUNE-1 or LSD1 with inhibitors AA7.1 and SP-2577, respectively. They conclude, combination of these two small molecules could be used in a second line treatment in advanced therapeutics against Gr3 MB. Inhibition lead to differentiation and microenvironment cytotoxicity, but also impaired oxphos phorylation through mitochondria.

Abstract

Q1) Generally well written. Describes study well.

R1-A1) We thank reviewer for his/her valuable positive discussion about our abstract, and study description.

Q2) Note, more commonly inhibition of oxphos is considered oncogenic such that the inhibition of OP here needs at least monitoring.

R1-A2) Indeed this comment is valuable for discussion, we address this in the Discussion section.

Introduction

Q3) Generally, a nice intro to MB and features of each grounds the reader well to the MB groups in general.

R1-A3). Many thanks for nice words about out intro.

Q4) The introduction is probably overlong with a little excess detail around LSD1 action/targeting in particular that could be pared a little and partially move to the discussion although clearly background and stage setting for PRUNE1 and LSD1 needs to occur to some extent.

R1-A4) We appreciate the reviewer’s valuable suggestion. As requested, we have removed this paragraph from the introduction: “LSD1/KDM1A was found negatively correlated with the expression of CD8+ on T cells and positively correlated with that of PD-L1. Shen et al. observed also that LSD1 inhibited the response of T cells in the TME of gastric cancer by inducing the accumulation of PD-L1 in exosomes. Recently, it has been shown that LSD1 inhibition in triple negative breast cancer cell lines induces expression of CD8+ T cell-attracting chemokines, including CCL5, CXCL9, and CXCL10. Soldi et al. investigated the ability of LSD1 pharmacological inhibition to promote anti-tumor immunity and T-cell infiltration in small cell carcinoma of the ovary hypercalcemic type (SCCOHT) and ovarian clear cell carcinomas (OCCC) cell lines. They found that SP-2577 stimulated IFN-dependent anti-tumor im-munity in SCCOHT and promoted the expression of PDL1 in both SCCOHT and OCCC. Recently Hong et al. reported that LSD1/KDM1A promotes NSCLC (non-small-cell lung cancer) metastasis through the TGF-β1/Smad pathway.” This information is already present in discussion.

Q4) From the authors earlier paper, it appeared that PRUNE-1 was over-expressed and PTEN consequently reduced in groups 3 and 4 MB accepting that some pathway components were higher in G3 v G4 eg NME-1. Do the authors feel that G4 MB may also benefit with possibly an extension study in the future, or do they feel it unlikely based on previous data?

R1-A4) We agree with the reviewer that it would be fascinating to explore G4 MBs if the same combinatorial approaches of targeting both LSD1/KDM1A and Prune1 show promising results. At this time, we are focusing on Gr3 treatment. It is not excluded that in future efforts we will address this important note.

Results

Q5) Figure 1B shows impressive detriment with KDM1A expression in Grp 3 and subtle in MB as a whole. This suggests there may not be a detriment at all with some types, suggest showing curves for all sub-types individually here to clarify the prognostic specificity for Gp3.

R1-A5) We thank reviewer for raising this issue. We have now analyzed and presented in Figure Supplemental 1A, 1B, 1C Kaplan-Meyer survival curves showing LSD1/KMD1A high or low expressed in all MB subgroups. The data show that in WNT and SHH subgroups the percentage of survival is worse when LSD1/KDM1A has a low expression, although not with a statistically relevance (p-values>0.05) (Figure Supplemental 1 A and B). Conversely, we found that in Gr4 there is trend similar to Gr3 (showed in Figure 1D but not with statistical significance (p-value =0.055). These observations were included in the text Results section (lines 176-180).

Q6) The Gp3 curves for high/high v low/low in 1D look less impressive than KDM1A alone in 1B – is PRUNE1 expression adding prognostic data to the model or is KDM1A alone just as good or better (clearly numbers are low).

R1-A6) Kaplan-Meyer survival curves for both PRUNE1 and LSD1/KDM1A showed in Gr3 MB a similar prognostic value and no significant different effect on the survival is observed (Figure 1I). For this reason, we investigated whether a high expression for both genes might have a worse prognostic value (Figure 1K).  We have now added this information in the Results section, now line 195-196.

Q7) The PRUNE1-KDM1A link is also underwhelming for all MB but looks solid for Grp 3, suggest show all individual types here also.

R1-A7) We thank the reviewer for his/her suggestion. We added the data for all MB subgroups (Figure Supplemental 1D, 1E, 1F). We observed that in WNT subgroup there is no statistical positive correlation between LSD1/KDM1A and PRUNE1 expression (Figure Supplemental 1D). Conversely in SHH and Gr4 subgroups we observed a statistically significant correlation between two genes (Figure Supplemental 1E and F). This is now commented in Resul section 189-190.

Q8) The epigenetic KDM1A chip data is in SH-SY5Y cells – are these MB Gp3 sub-type? If so, was a difference seen in an example of another sub-type considering the lower correlation outside Gp3 MB?

R1-A8) We thank reviewer for this comment. This is a neuroblastoma (NB) cell line. More in detail, it is a subline of the neuroblastoma cell line SK-N-SH, which was established from a metastatic bone tumor from a 4-year-old cancer patient. SH-SY5Y is a cellular model usually employed for neuronal development studies. ChIP-Seq data in this cell line have been used to generalize the epigenetic regulation of LSD1/KDM1A on the target PRUNE-1.

Q9) Was the inhibitory activity of LSD1 blockade also tested on other MB types?

R1-A9) We thank the reviewer to raise this important point. To please his/her request we have now performed a combinational treatment with both inhibitors in MB cell line of SHH subgroup (DAOY cells) and Gr3/4 (D283-Med cells) (see Figure Supplemental 3F). We discussed these results in lines 273-276.

Q10) The fall in OTX2 appears more prominent than that of PRUNE1 in Fig 2D. Do you feel the former is still due to the latter?

R1-Q10) Considering the ChIP-Seq data in Figure Supplemental 2C and D showing the presence of LSD1/KDM1A on OTX2 genomic region, we could speculate that OTX2 might be regulated both in direct way (through a direct epigenetic regulation on target) and indirect way (through PRUNE-1 regulation). Thus, this could be the reason for a fall in OTX2 more prominent than in PRUNE-1 with the treatment of both inhibitors (PRUNE-1 inhibitor and LSD1/KDM1A inhibitor). This hypothesis is included in Results section lines 242-245.

Q11) Did you test the impact of the two inhibitors on the primary MB cells or just look at protein expression changes?

R1-A11). In response to reviewer request both RNA levels (through RNA-Seq) and protein levels (through Western Blot) of the targets of main signaling pathway of Gr3 MB were analyzed (Figure 4, Figure 5 and Supp Figure 5).

Q12) The reactivation of differentiating pathways and cytotoxic immune pathways is nice and fits with benefit.

R1-A12) We thank reviewer for his/her positive comments.

Q13) I was a little surprised that mitochondrial gene expression patterns suggested an inhibition of oxphos. It is also slightly concerning as malignant cells tend to channel energy metabolism through glycolysis (ie the Warburg effect as the authors reference) presumptively to repurpose Acetyl CoA into other pathways and use less oxphos with oxphos upreg usually being associated with benefits.

R1-A13) We thank the reviewer for the opportunity to comment and clarify this point. Our RNA-Seq data show (Figure 5A and B) that several genes involved in OXPHOS are significantly negatively regulated upon inhibitors treatment. The role of metabolism and the weight of aerobic glycolysis or mitochondrial respiration in cancer cells is still debated. However, several studies have been shown that not all tumors share these properties of aerobic glycolysis. Glycolysis in most cancer is promoted by oncogenic driver mutations not by mutations in genes whose activities are related mitochondrial respiration complexes (Vander Heiden et al., 2009). Then inactivation of the mitochondrial transcription factor “TFAM” depletes mitochondria from tumor cells and impairs K-ras lung tumor (Weinberg et al., 2010). Depleting mitochondrial DNA from tumor cells by poisoning mitochondrial DNA replication compromises tumorigenesis (Tan et al., 2015). Furthermore, in Drosophila brain tumors, inhibition of OXPHOS through knockdown of mitochondrial complex I or V caused a decrease in tumor growth and heterogeneity (Van den Ameele et al., 2019). In brain tumors, Kuramoto et al. (2020) found higher expression levels of mitochondrial transcription factor TFAM in undifferentiated glioma stem cells (GSC) compared to more differentiated glioma cells expressing GFAP (Kuramoto et al., 2020). Interestingly, the expression levels of cytochrome b, a component of mitochondrial complex III, as well as cytochrome c oxidase (COX) 2 and COX3, components of complex IV, were markedly reduced after cell differentiation (Kuramoto et al., 2020). Of importance, evidence in the literature suggests that metastatic cells in the brain upregulate OXPHOS-related genes (Faubert et al., 2020). According to this, Martirosian et al. (2021) observed that MB cells cultured in artificial cerebrospinal fluid (aCSF) display high levels of OXPHOS (Martirosian et al., 2021). Of note, Young et al. (2023) used a mouse model of metastatic triple negative breast cancer (TNBC) to dissect the metabolic profile of metastasis-initiating cells (MICs) and uncovered that MICs relied on fatty acid oxidation (FAO) and the oxidative tricarboxylic acid (TCA) cycle and that these contributed to mitochondrial acetyl-CoA generation (Young et al., 2023). Having Gr3 MB the highest metastatic potential among MB, we are convinced that Gr3 MB cells displayed high rate of OXPHOS allowing them to initiate dissemination and metastasis.  Part of the following considerations are added in discussion section, lines 508-531

Discussion

Q14) When cells differentiate it can impact response to other treatment components eg chemo or RT, possibly with reduced proliferation reducing sensitivity, but also as a counterpoint, loss of EMT and loss of stem cell characteristics can also increase sensitivity. I presume combinations have not yet been done but bears discussion.

R1-A14) We thank the reviewer for this consideration. We now added the expression of N-cadherin (marker of active EMT) and we observed a reduction in the levels of this protein in D425-med cells treated with AA7.1 and with SP-2577 with a greater effect when the combination of two drugs was used. These data are presented now in Figure Supplemental 3E and Result section, lines 269-271.

Q15) Agree that discussion regarding the degree to which this may be a group 3 restricted benefit is warranted as Gp4 appeared to have some similar changes from authors previous paper.

R1-A15) We thank reviewer for his/her positive comment, these were now included in Discussion section line 413-416.

Q16) The discussion around modulation of components of the TME is warranted as this could be a significant strength of the therapy bearing in mind MBs low immunogenicity generally.

R1-A16) We thank reviewer for his/her comment. Although MB is reported as a low immunogenicity cancer, there are growing evidence about the effects of immune system in the TME also for brain cancers as we reported in discussion. We agree that this could be of significant additive strength for MB therapies.

Q17) Regarding the mitochondrial impacts, the authors do reference increased OXPHOS in some cancers cf normal such that a treatment inhibiting OXPHOS could be beneficial but this is not, I feel, where the weight of published literature sits across cancers. There could be an MB-specific effect here as they suggest. Could mention that dissecting out this effect by specific OXPHOS inhibition to make sure this isn’t detrimental could be considered in the future.

R1-A17) We thanks reviewer to raise this point. As request, here we list several examples where inhibitors of mitochondrial respiration complexes were used for therapeutic issues in MB. Gr3 MB tumors are more resistant to chemotherapy (Linke et al., 2021). Linke et al. (2023) observed that Gr3 MB tumor model was characterized by multiple subpopulations with greatly enhanced OXPHOS and tricarboxylic acid (TCA) cycle activity with very high levels of fumarate (Linke et al., 2023). While vincristine alone was not sufficient to decrease cell viability of Gr3 MB tumors, combination with the NRF2- a fumarate-mediated oxidative stress pathway member- inhibitor significantly enhanced the chemotherapy effect (Linke et al., 2023). Contenti et al. (2023) reported that the use of Phenformin, a mitochondrial complex 1 inhibitor, induced significant cell death in Gr3 MB cells (Contenti et al., 2023). Martell et al. (2023) noted that suppressing OXPHOS by using complex-I inhibitors (Phenformin, Rotenone, IACS-010759), impaired the cell number of HD-MB03 (MYC-amplified Gr3 MB cells) after 24 hours of treatment (Martell et al., 2023). In particular, IACS-010759 treatment promotes differentiation and suppresses growth and stemness of Gr3 MB cells (Martell et al., 2023). Of note, oral administration of IACS-010759 impairs tumor growth and prolongs survival in a pre-clinical orthotopic Gr3 MB xenograft model (Martell et al., 2023). Rossi et al. (2022) found that b-blockers are able to disrupt mitochondrial bioenergetics increasing radiotherapy efficacy in MB (Rossi et al., 2022). More in detail, response to b-blockers is associated with inhibition of energy metabolism in MB cells (Rossi et al., 2022). By measuring the mitochondrial respiration via the oxygen consumption rate (OCR) and the glycolytic activity via the extracellular acidification rate (ECAR), resulted that the b-blockers treatment induced a significant drop in mitochondrial respiratory functions in ONS-76, HD-MB03, UW228-2 and DAOY cells and a strong reduction in ATP production in all these MB cell lines (Rossi et al., 2022). Part of these data presented in literature are added in discussion section, lines 542-558. 

Q18) The mutation of mitochondrial acyl-CoA ligase in this case is interesting. Raises the question as to whether the mitochondrial effects would be seen if a comparator wild type case was analyzed.

R1-A18) In our WES data we observed the mutation of gene ACSF2. This gene encodes for a mitochondrial enzyme member of acyl-CoA synthetase family, implicated in fatty acid oxidation, a lipid metabolic pathway that breaks down fatty acids into smaller acyl-CoA molecules (Nakahara et al., 2012). The acyl-CoA enters the mitochondria and undergoes a progressive cleavage into two carbon acetyl-CoA molecules, producing energy molecules FADH2 and NADH that participate in the TCA (Dhiman et al., 2019). The acyl-CoA enters the mitochondria and undergoes a progressive cleavage into two carbon acetyl-CoA molecules, producing energy molecules FADH2 and NADH that participate in the TCA (Dhiman et al., 2019). Acetyl-CoA generated in mitochondria condenses with oxaloacetate to produce citrate, which is oxidized in the TCA cycle, enabling ATP production through OXPHOS (Sivanand et al., 2018). We supposed this enzyme cooperates to sustain mitochondrial metabolism and probably its aberrant expression of its mutant form has a more pathogenic potential. Interestingly, the gene ACSF2 was reported as a biomarker for prognosis of breast cancer and colon adenocarcinoma (Wang et al., 2021, Parsazad et al., 2023). Its expression level of ACSF2 was found higher in acute myeloid leukemia ovarian cancer tissues than in normal tissues (Song et al., 2021, Wang et al., 2022). Of note, by interrogating the dataset Pediatric Brain Cancer (CPTAC/CHOP, Cell 2020, https://www.cbioportal.org/), we observed that for the gene ACSF2 is reported the gain of copy-number in most MB samples taken into consideration. These considerations are now discussed at lines 570-589.

Reviewer 2 Report

Comments and Suggestions for Authors

The authors identify and characterize the effects of inhibiting potential new epigenetic protein targets using their inhibitors, either alone or in combination, aiming for tailored therapies for medulloblastoma. Molecular techniques and database querying are employed. The authors demonstrate the impact of these inhibitors on molecular pathways associated with differentiation, tumor immunity, oxphos metabolism, and a cytological parameter, namely the cell index. The study is well-executed, well-written, and valuable; however, there are several major and minor points to address:

Major points:

1) The cell index solely indicates an increase in cell numbers but does not provide insight into the contribution of cell death, which should be acknowledged.

2) There is a lack of direct cytological evidence showing the effect of inhibitor treatment on oxphos and differentiation, both in cell lines and primary lines.

Minor points:

1) It is acknowledged that quantitative results from western blots, particularly with chemiluminescence, may be somewhat questionable. Although analysis of whole gels can be convincing, the gels presented in the main figures are less so. Could the authors provide more compelling gels, particularly for Prune-1 in figure 2, which appears less decreased in the double treatment compared to p-smad-2 in the same figure (especially in comparison with beta-actin), and for Prune-1 and OTX2 in figure 4?

2) There is some redundancy in the discussion; for instance, the explanation of antigen presentation could be omitted to streamline the text.

3) The results display the surgical procedure to confirm that the patient from whom the primary cells were obtained indeed had medulloblastoma, but the images and description could be minimized.

4) The necessity to describe postoperative therapy for research purposes is unclear.

5) Certain figures are nearly illegible: immunohistochemistry in figure 3D, the volcano plot in figure 4E, and figure 5.

6) The platform used to calculate the cell index must be indicated in the materials and methods section

Overall, the study can be accepted if the discussion emphasizes the requirement for additional direct evidence in future studies, despite the abundance of convincing genetic, expression, and molecular data.

Comments on the Quality of English Language

Correct some spelling errors (e.g., "tumoral tissue" instead of "tumor tissue").

Author Response

Reviewer 2

Comments and Suggestions for Authors

The authors identify and characterize the effects of inhibiting potential new epigenetic protein targets using their inhibitors, either alone or in combination, aiming for tailored therapies for medulloblastoma. Molecular techniques and database querying are employed. The authors demonstrate the impact of these inhibitors on molecular pathways associated with differentiation, tumor immunity, oxphos metabolism, and a cytological parameter, namely the cell index. The study is well-executed, well-written, and valuable; however, there are several major and minor points to address:

Major points:

Q1) The cell index solely indicates an increase in cell numbers but does not provide insight into the contribution of cell death, which should be acknowledged.

R2-A1) This is an important and relevant consideration and we thank the reviewer. Following his/her request, we have analyzed the activated caspase 3 as marker of induction of cell death upon individual and combinatory treatment with our drugs. Data obtained through WB showed that there is no caspase activation following these treatments in D425- Med cells. Data are presented now in Figure 2D and in Results section, lines 260-263. 

Q2) There is a lack of direct cytological evidence showing the effect of inhibitor treatment on oxphos and differentiation, both in cell lines and primary lines.

R2-A2) We thank the reviewer for his/her critical comments. For this reason, in order to support our RNA-seq results showing up-regulation of genes involved in neuronal differentiation, we observed in particular the expression levels of differentiation marker TUBB3. It is higher expressed in primary MB Gr3 cells treated with our drugs compared to vehicle cells (see Figure Supplemental 5C). Furthermore, we added an immunofluorescence analysis using antibody against another marker of differentiation, the Glial Fibrillary Acid (GFAP). The images are presented in Figure 4D and Results section, lines 338-348. The treatment of D425-Med cells with the combination of AA7.1 and SP-2577 resulted in a significant increased expression of GFAP. Thus, this demonstrates a commitment of treated cells toward a more differentiated phenotype.

Minor points:

Q3) It is acknowledged that quantitative results from western blots, particularly with chemiluminescence, may be somewhat questionable. Although analysis of whole gels can be convincing, the gels presented in the main figures are less so. Could the authors provide more compelling gels, particularly for Prune-1 in figure 2, which appears less decreased in the double treatment compared to p-smad-2 in the same figure (especially in comparison with beta-actin), and for Prune-1 and OTX2 in figure 4?

R2-A3) We appreciate the reviewer’s suggestion. Following his/her request we have replaced more convincing gel images for PRUNE1 in figure 2 and figure 4 and OTX2 in Figure 4.

Q4) There is some redundancy in the discussion; for instance, the explanation of antigen presentation could be omitted to streamline the text.

R2-A4) We appreciate the reviewer’s suggestion. Accordingly, we removed this part from the revisited manuscript. Were removed the following paragraph: “The process of antigen presentation begins with the cancer cells expressing proteins on their surface, which are then recognized by specialized immune cells called antigen presenting cells (APCs). These APCs, such as dendritic cells, then internalize the cancer cell proteins and degrade them into smaller peptides displayed on the surface of the APC, along with proteins of major histocompatibility complex (MHC) molecule.”

Q5) The results display the surgical procedure to confirm that the patient from whom the primary cells were obtained indeed had medulloblastoma, but the images and description could be minimized.

R2-A5) We thank the reviewer for these comments. Accordingly, we removed Figures 3B and 3C, moved in Figure Supplemental 4A and 4B. In addition, the following paragraph was removed in the final revision manuscript: “The intraventricular mass shows vivid contrast enhancement, increased perfusion parameters and cellular turnover. Images show the slightly ex-vacuo enlargement of the fourth ventricle and the resolution of compression and displacement effects.”

Q6) The necessity to describe postoperative therapy for research purposes is unclear.

R2-A6) We thank the reviewer for this consideration. We have removed the description of post-surgical therapies presented in Materials and Methods section.

Q7) Certain figures are nearly illegible: immunohistochemistry in figure 3D, the volcano plot in figure 4E, and figure 5.

R2-A7) We thank the reviewer for his/her observations. We have improved the quality of all the images in the revised version.

Q8) The platform used to calculate the cell index must be indicated in the materials and methods section.

R2-A8) We added this information in the section dedicated to Cell Index Assay presented in Materials and Methods Section.

Q9) Overall, the study can be accepted if the discussion emphasizes the requirement for additional direct evidence in future studies, despite the abundance of convincing genetic, expression, and molecular data.

R2-A9) We thank the review for his/her suggestion. We added in the manuscript now in discussion section. A better dissection of the molecular mechanisms driving Gr3 MB, of the effects of anti-Prune-1 molecule AA7.1 and LSD1 inhibitor SP-2577 and in vivo validation of our findings in a mouse model of Gr3 MB is still required.  Although this, we think that an integrated multidimensional diagnostic approach involving genomic mutational signatures and transcriptomic features could be helpful to identify potential actional targets and stratify the patients for treatments. An integrated approach is important to understand the different responses to the therapies. In conclusion, we would like to propose the combination of two inhibitory molecules (AA7.1 and SP-2577) for treatment of patients affected for Gr3 MB for which efficient therapies are current lacking.

Comments on the Quality of English Language

Correct some spelling errors (e.g., "tumoral tissue" instead of "tumor tissue").

We thank reviewer for his/her comments. Following his/her suggestion, we have corrected errors as requested.

Round 2

Reviewer 2 Report

Comments and Suggestions for Authors

The reviewer appreciates the effort made by the authors in meeting the requirements and now considers the manuscript suitable for publication.